# Epithelial GREMLIN1 disrupts intestinal epithelial-mesenchymal crosstalk to induce a wnt-dependent ectopic stem cell niche through stromal remodelling

Eoghan J. Mulholland [1,10], Hayley L. Belnoue-Davis [1,10] ✉,
Gabriel N. Valbuena[1], Nuray Gunduz[2], Amelia Ligeza[1], Muyang Lin[1], Sujata Biswas[1],
Ester Gil Vasquez[1], Sulochana Omwenga[1], Nadia Nasreddin[1],
Michael C. Hodder[2,3], Lai Mun Wang[4], Aik Seng Ng [5], Elizabeth Jennings[6],
Kim S. Midwood [6], Neesha Dedi [7], Shazia Irshad [1,5], Rachel A. Ridgway [2],
Toby J. Phesse [5,8], James East [9], Ian PM Tomlinson [5], Gareth CG Davies [7],
Owen J. Sansom [2,3] & Simon J. Leedham [1,9]

In homeostasis, counterbalanced morphogen signalling gradients along the vertical axis of the intestinal mucosa regulate the fate and function of epithelial and stromal cell compartments. Here, we use a disease-positioned mouse and human tissue to explore the consequences of pathological BMP signalling dysregulation on epithelial-mesenchymal interaction. Aberrant pan-epithelial expression of the secreted BMP antagonist Grem1 results in ectopic crypt formation, with lineage tracing demonstrating the presence of Lgr5(−) stem/progenitor cells. Isolated epithelial cell Grem1 expression has no effect on individual cell fate, indicating an intercompartmental impact of mucosal-wide BMP antagonism. Treatment with an anti-Grem1 antibody abrogates the polyposis phenotype, and triangulation of specific pathway inhibitors defines a pathological sequence of events, with Wnt-ligand-dependent ectopic stem cell niches forming through stromal remodelling following BMP disruption. These data support an emerging co-evolutionary model of intestinal cell compartmentalisation based on bidirectional regulation of epithelial-mesenchymal cell fate and function.

The unique, crypt-based architecture of the gut mucosa has enabled the study of adult stem cells and daughter cell fate determination. In 2007, the Clevers group unequivocally demonstrated multipotency and self-renewal in a population of *Leucine rich repeat containing G protein-coupled receptor 5 (Lgr5)* expressing crypt-base columnar cells (CBC)[1]. However, these cells were subsequently found to be dispensable for tissue regeneration[2] and multiple putative populations of *Lgr5(−)* stem cells have been described in regenerative and

[1]Centre for Human Genetics, Roosevelt Drive, University of Oxford, Oxford, UK. [2]Cancer Research UK Scotland Centre, Glasgow, UK. [3]Institute of Cancer Sciences, University of Glasgow, Glasgow, UK. [4]Department of Laboratory Medicine, Changi General Hospital, Singapore, Singapore. [5]Department of Oncology, University of Oxford, Oxford, UK. [6]Kennedy Institute of Rheumatology, Nuffield Department of Orthopaedics, Rheumatology and Musculoskeletal Science University of Oxford, Oxford, UK. [7]UK Research, UCB Pharma, Slough, Berkshire, UK. [8]The European Cancer Stem Cell Research Institute, School of Biosciences, Cardiff University, Cardiff, UK. [9]Translational Gastroenterology Unit, John Radcliffe Hospital, University of Oxford, and Oxford NIHR Biomedical Research Centre, Oxford, UK. [10]These authors contributed equally: Eoghan J. Mulholland, Hayley L. Belnoue-Davis. ✉e-mail: hayley.davis@well.ox.ac.uk

pathological settings[3,4]. Advances in fate mapping technologies continue to iterate the intestinal stem cell model, with two recent studies identifying a spatially and transcriptionally distinct population of upper crypt *Fibroblast growth factor binding protein 1 (Fgfbp1)* expressing cells, capable of reconstituting lost *Lgr5(+)* cells and the epithelium following damage[5,6].

Luminally directed migration of stem cell progeny results in continual epithelial turnover along a crypt-to-villus cell escalator[1]. As daughter cells migrate along this axis, stem-to-differentiated functional segregation is regulated by the establishment of opposing gradients of intercompartmental morphogenetic signalling[2,3]. From the crypt isthmus upwards to the luminal surface, stromally secreted Bone Morphogenetic Protein (BMP) ligands mediate cell differentiation and apoptosis, whereas in the crypt base, restricted Wnt signalling drives crypt base columnar cell stemness, transit amplifying cell division and regulates progenitor cell fate. Exclusive sub and peri-crypt stromal cell expression of secreted, ligand-sequestering BMP antagonists (BMPi) protects against premature differentiation of stem cell populations[4–6]. Developmentally, these polarised signalling gradients are both a cause and consequence of the distinct architecture of the intestinal mucosa, and the inter-relationship of the epithelial and stromal cell compartments. In the developing small intestine, villus formation precedes crypt morphogenesis. Physical buckling of the epithelial surface generates projecting villi that results in the formation of a sub-epithelial, BMP-expressing stromal cell aggregation, known as the villus cluster[7]. This pro-differentiation signalling at the luminal surface repels putative stem cells to the base of the villus where developing crypts generate restricted, wnt-high crypt basal stem cell niches[7]. The impact of polarised signalling gradients in regulating homeostatic epithelial cell fate is established but recent findings also demonstrate the morphogenetic regulation of stromal cell function along the crypt-villus axis. Kraiczy et al., used cell marker expression to segregate intestinal fibroblast cell populations into distinct functional subsets[8]. Mesenchymal expression of stem cell supporting signalling factors, such as R-Spondins and BMPi, were restricted to subcryptal Cd81(+) Pdgfra(lo) trophocytes, and pericryptal Cd81(−) Cd55(+) Pdgfra(lo) stromal cells, resulting in the maintenance of a discrete crypt stem cell niche. The authors proposed that BMP ligand signalling arising from luminal aggregates of Pdgfra(hi) sub-epithelial myofibroblasts (SEMF) regulates this variable stromal cell activity, suppressing stem cell niche supporting functionality at the luminal surface[8]. Consequently, intercompartmental secreted BMP signalling, generates a polarised signalling gradient along the intestinal vertical axis, that regulates not only epithelial cell fate but also stromal cell function to restrict appropriate stem cell niche activity exclusively to the crypt base.

This interdependence between mucosal structure and function regulated by secreted signalling pathways, renders the intestine susceptible to disruption of the morphogen gradient balance. The majority of intestinal lesions arise through epithelial mutation-induced activation of the wnt pathway, however, pathological inactivation of BMP signalling is also capable of initiating human polyp formation. This is epitomised by the human polyposis syndromes Juvenile Polyposis and Hereditary Mixed Polyposis Syndrome (HMPS), which arise through germline mutations in the BMP pathway. In HMPS, a duplication mutation on chromosome 15 causes ectopic pan-epithelial expression of the secreted BMPi, *GREMLIN1*, which is normally restricted to exclusive expression by sub and peri crypt stromal cells[9]. In mouse models, forced epithelial expression of secreted BMPi, such as *Noggin* or *Grem1* leads to architectural distortion with the generation of ectopic crypts[10–12]. These ectopic crypt foci (ECF) are populated by Lgr5(−) progenitor cells, that proliferate and acquire oncogenic driver mutations, with eventual tumorigenesis arising from a cell-of-origin situated outside of the regulatory confines of the crypt base[12].

In this work, we use a disease-positioned animal model of HMPS to mechanistically explore the consequences of pathological morphogen gradient dysregulation on epithelial and mesenchymal cell compartment structure and function. We investigate the establishment and regulation of ectopic stem cell niches and demonstrate the impact of therapeutic intervention on the intestinal neoplasia initiated by disruption of polarised BMP signalling gradients.

## Results

### The developing *Vil1-Grem1* intestine is phenotypically unaffected by embryonic epithelial *Grem1* expression

In Hereditary Mixed Polyposis Syndrome (HMPS) patients the switch from mesenchymal to epithelial *GREM1* expression arises from a germline duplication mutation on chromosome 15[7], possibly through amplification of upstream enhancer regions, including known CRC predisposition SNP's[8]. Although it is an autosomal dominant condition, the polyposis phenotype predominantly develops in adults, with clinical presentation at a median age of 40[9]. Our previous work had generated a disease-positioned mouse model of HMPS called *Vil1-Grem1* using the *Villin1* promoter to drive aberrant intestinal epithelial expression of the secreted BMP antagonist *Grem1*. This induces a pronounced murine pan-intestinal polyposis in adult mice (median survival 250 days) with lesions that phenocopy the mixed polyp histology of the human condition, including the formation of ectopic crypts, best seen within the villus in small intestinal lesions[12]. Although the mouse phenotype only becomes clinically evident in adult animals, the *Villin1* promoter that drives aberrant intestinal epithelial expression is active in the mouse embryo from the time of axial rotation[10], so we undertook timed matings of *Vil1-Grem1* animals to assess embryonic and post-natal intestinal phenotype. *Villin1* expression is evident in the developing intestinal epithelium but despite this, ectopic *Grem1* expression is comparatively low, and is predominantly appropriately restricted to the stromal cell compartment (Fig. S1A). As development proceeds into postnatal animals, aberrant epithelial *Grem1* expression is increasingly evident, however this does not have a major impact on newborn intestinal epithelial architecture, with no evidence of ectopic crypts and appropriate restriction of cells expressing stem cell markers to the nascent crypts. In postnatal to adult tissue development (P10 to 100 days) the effect of ectopic epithelial *Grem1* expression became evident with the emergence of enlarged villi, ectopic crypts and early polyps. Concomitant with this morphological change there was an incremental reduction in the proportion of differentiating, CK20(+), cells and a resulting steady expansion in crypt-villus unit cell number and overall size of the adult mouse intestine[12] (Fig. S1B–D). These data show that embryological activation of aberrant epithelial *Grem1* expression does not appear to induce an immediate postnatal polyposis phenotype, but the subsequent derangement of appropriate cell differentiation and shedding expands the intestinal epithelial cell population with epithelial folding and the gradual development of invaginating ectopic crypts, with subsequent polyposis which emerges in adult animals (Fig. S1C, D).

### Lineage tracing from ectopic crypt cells

Within ectopic crypt cells in adult animals we have previously shown aberrant cell proliferation but notable lack of expression of the crypt base columnar stem cell marker *Lgr5*[12]. In light of the recently published upper crypt stem cell model[5,6], we examined expression of the *Fgfbp1* marker, and were able to show extensive aberrant expression within the pathological progenitor cells situated in the ectopic crypt in *Vil1-Grem1* animals (Fig. 1A). There was overlap of expression with more established stem/progenitor cell markers, such as *Sox9* in the villus ectopic crypts, however, all of these markers were also expressed in the homeostatic progenitor cell population in the normal crypts (Figs. 1A and S1E). In order to functionally demonstrate the stem cell potential of ectopic crypt foci, we generated *Sox9-CreER^{T2}; Rosa26^{YFP};*

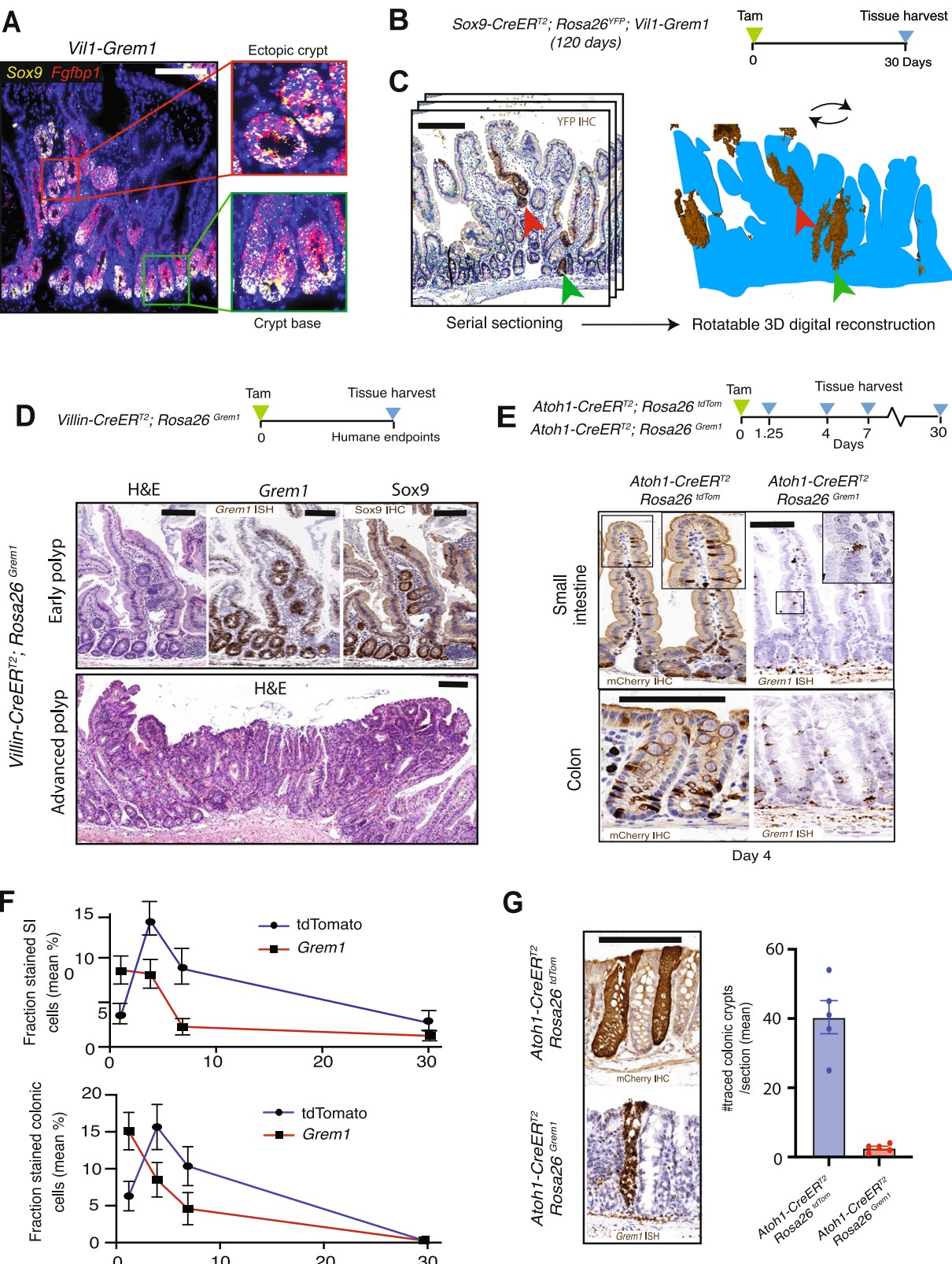

**F**

**G**

*Vil1-Grem1* mice to assess whether ectopic crypt *Sox9*(+) cells were capable of generating lineage tracing ribbons in vivo. We aged animals for 120 days to establish a pan-intestinal polyposis phenotype and then induced recombination with a single pulse of tamoxifen to activate YFP expression in *Sox9*(+) cells (Fig. 1B). Recombined animals were sacrificed after 30 days and chromogenic immunohistochemistry against YFP was used to identify lineage traced cell populations arising from

*Sox9* (+) cells situated within both the normal and ectopic crypt bases. Although, we saw frequent lineage tracing ribbons arising above the crypt isthmus in small intestinal polyps, we recognised that in two dimensional sections, these visualised traced cells could conceivably have arisen from proximally situated *Sox9*+ cell recombination, within the normal crypt base. To exclude this, we undertook whole lesion serial sectioning and used tissue alignment and digital reconstruction

**Fig. 1 | Ectopic crypt lineage tracing and secretory cell fate. A** Fluorescent co-ISH staining of *Sox9* and *Fgfbp1* in 120 day *Vil1-Grem1* polyp shows co-localised staining both in normal crypt base (green box) and ectopic crypt (red box) (*n* = 5 mice). **B** Schematic shows recombination and harvesting timepoint for *Sox9-CreER^T2^; Rosa26^YFP^;Vil1-Grem1* mouse model. **C** Tissue serial sectioning and lineage tracing ribbon identification with anti-YFP immunohistochemistry (brown), followed by tissue alignment and polyp 3D reconstruction of lineage tracing ribbons (brown) (*n* = 1 whole mouse SI). **D** H&E, Grem1 and Sox9 staining of Villin-CreER^T2^; Rosa26^Grem1^ early and advanced polyps (*n* = 5). **E** Secretory cell staining in small intestine and colon 4 days post-recombination in *Atoh1-CreER^T2^,Rosa26^tdTom^* or *Atoh1-CreER^T2^; Rosa26^Grem1^* mice stained with mCherry IHC or *Grem1* ISH respectively. **F** Number of mCherry IHC (blue lines) or *Grem1* ISH (red lines) stained small intestinal or colonic epithelial cells over time following recombination (*n* = 5 mice per group). Data were ±s.e.m. **G** Quantification of number of fully lineage traced colonic crypts across gut roll sections at day 30 post recombination in *Atoh1-CreER^T2^,Rosa26^tdTom^* or *Atoh1-CreER^T2^; Rosa26^Grem1^* mice stained with mCherry IHC or *Grem1* ISH respectively (*n* = 5 mice per genotype, two-tailed *t* test, *p* value = 0.0013). Data were ±s.e.m. Scale bar 200 μm throughout. Source data are provided as a Source Data file.

of individual polyps (HeteroGenius, Leeds, UK) to examine the three-dimensional path of lineage traced cells (Supplementary movies 1 and 2). This allowed us to map tracing ribbons in their entirety and track the path of YFP labelled cell progeny (Fig. S2A, Movie 1,2). Using this technique were able to identify tracing ribbons, originating above the crypt base from villus ectopic crypts, that were spatially distinct from traces emerging from neighbouring crypt basal cells (Fig. 1C). Together this work demonstrated that proliferating *Sox9* (+) cells situated in the ectopic crypts within the villus of *Vil1-Grem1* mice are capable of generating multicellular lineage tracing ribbons, and thus show functional stem cell potential in vivo.

### Aberrant *Grem1* expression in individual progenitor cells does not alter cell fate trajectory

In homeostasis, *Grem1* expression from sub and peri-crypt stromal cells is an important constituent of the niche, maintaining crypt basal *Lgr5(+)* stem cell number[5]. Aberrant panepithelial expression promotes ectopic stem cell behaviour in the germline *Vil1-Grem1* model of HMPS, however, this model is unable to dissect whether aberrant autocrine epithelial Grem1 expression intrinsically promotes stemness in cells situated outside of the crypt basal niche. To examine this we generated an inducible *Lox-STOP-Lox;Rosa26^Grem1^* model (hereafter *Rosa26^Grem1^*). Initially we crossed it with *Villin-CreER^T2^* to assess the impact of pan-epithelial activation of *Grem1* in adult mice. Although the resultant phenotype was less profound than in the germline *Vil1-Grem1* model, all *Villin-CreER^T2^;Rosa26^Grem1^* animals developed polyposis with the initial establishment of ectopic crypts on the villi containing aberrantly proliferating *Sox9(+)* stem/progenitor cells (Fig. 1D). Next, we used *Atoh1-CreER^T2^* to induce expression of either tdTomato marker (*Atoh1-CreER^T2^;Rosa26^tdTom^*) or *Grem1* (*Atoh1-CreER^T2^;Rosa26^Grem1^*) in individual *Atoh1*(+) secretory progenitor cells in steady state conditions (Fig. 1E). We then tracked individual cell fate through expression of either the active morphogen *Grem1* (with ISH), or the functionally neutral tdTomato marker (with IHC), taking care to quantify and contrast spatio-temporal fate of cells within animal groups when utilising these different methodological cell marking techniques (Fig. 1E). In the small intestine, expression of neutral tdTomato marker was seen in crypt basal cells within 30 h and was retained in small numbers of long-lived Paneth cells for 30 days. This pattern of Paneth cell fate trajectory was mirrored by individual cells expressing autocrine *Grem1*. Other tdTomato or *Grem1* expressing cells differentiated into goblet cells, moved along the crypt-villus axis and were appropriately lost over time through cell shedding. This loss was initially accelerated in individual cells aberrantly expressing *Grem1* (Fig. 1F). In the colon, occasional crypt lineage tracing could be seen from *Atoh1* derived cells, as a consequence of cell de-differentiation to *Lgr5(+)* CBC[13] however, consistent with accelerated cell loss, the number of lineage traced crypts was significantly reduced in cells expressing autocrine *Grem1* (Fig. 1G).

Together these data show that pan-epithelial expression of *Grem1* in adult mice, with concomitant disruption of mucosal-wide BMP signalling gradients, led to the emergence of an ectopic crypt stem/progenitor cell population similar to that seen in the disease-positioned germline *Vil1-Grem1* model. However, this is not the consequence of an intrinsic effect on individual cells as cell-autonomous *Grem1* expression in secretory progenitors did not enhance cell functional stemness, generate ectopic crypts or prevent appropriate terminal differentiation of that particular cell.

### Phenotype reversal through *Grem1* inhibition

The pathognomonic mixed polyposis phenotype in HMPS[9,14] and the *Vil1-Grem1* model is caused by aberrant epithelial expression of *Gremlin1*[12]. To see if we could reverse the effects of this secreted antagonist, we treated *Vil1-Grem1* animals with an anti-Grem1 antibody (UCB Ab7326 mIgG, UCB Pharma) which blocks sequestering of BMP ligands[15]. Long-term treatment with preventative intent was initiated either at weaning (35 days old) or in animals with an established polyposis (120 days old) (Fig. 2A). Treatment with UCB Ab7326 profoundly increased mean mouse survival in both treatment groups, with mice treated from weaning succumbing to old age, rather than from the burden of intestinal disease (Fig. 2B). Interestingly, antibody treatment of wildtype mice over the same length of time, reduced expression of crypt basal stem cell markers but had no impact on mouse phenotype (Fig. S2B).

Next, we undertook short-term dosing of 120-day-old mice with treatment intent, sacrificing animals at different weekly timepoints to assess the acute effect of the antibody on established polyps (Fig. 2C). This resulted in a remarkable reversion of the characteristic *Vil1-Grem1* pan-intestinal polyposis phenotype (Figs. 2D and S2C). Macroscopically >4 weeks of antibody treatment significantly reduced the size of the pathologically enlarged and thickened intestinal mucosa seen in these animals (Fig. 2D, E). Microscopically, there was re-imposition of recognisable villus architecture, with restoration of normal villus patterns of the CK20 differentiation cell marker. Antibody treatment led to a rapid and profound abrogation of polyposis, near elimination of villus ectopic crypts and absence of villus proliferating Ki67(+), Sox9(+), and *Fgfbp1(+)* cells. Olfm4, EphB2 and lysozyme expressing cells were once again appropriately confined to the base of the intestinal crypts in treated animals (Fig. 2F).

Next, we undertook single-cell RNA-sequencing analysis on wild-type, 10-week vehicle and antibody-treated *Vil1-Grem1* animals, utilising crypt and villus dissection to enable spatial compartmentalisation (Fig. 3). Single cell analysis confirmed the expansion of an aberrant proliferating stem/progenitor cell population on the villus marked by *Mki-67*, *Sox9* and *Fgfbp1* expression, and demonstrated the reversion of this population following antibody treatment. As BMP signalling has been shown to control enterocyte and secretory cell zonation[11], we assessed secretory cell fate in the single cell data and demonstrated the presence of aberrant co-expressing *Lyz1* and *Muc2* cells within the ectopic crypts in *Vil1-Grem1* animals (Figs. 2F and S3E). These cells were no longer seen following antibody treatment, indicating the reversible impact of disrupted pan-mucosal BMP gradients on secretory progenitor differentiation.

Together, these data show that aberrant epithelial *Grem1* expression induces reversible intestinal architectural change and that the mixed polyposis phenotype can be both prevented and reversed through sequestering inhibition of Grem1.

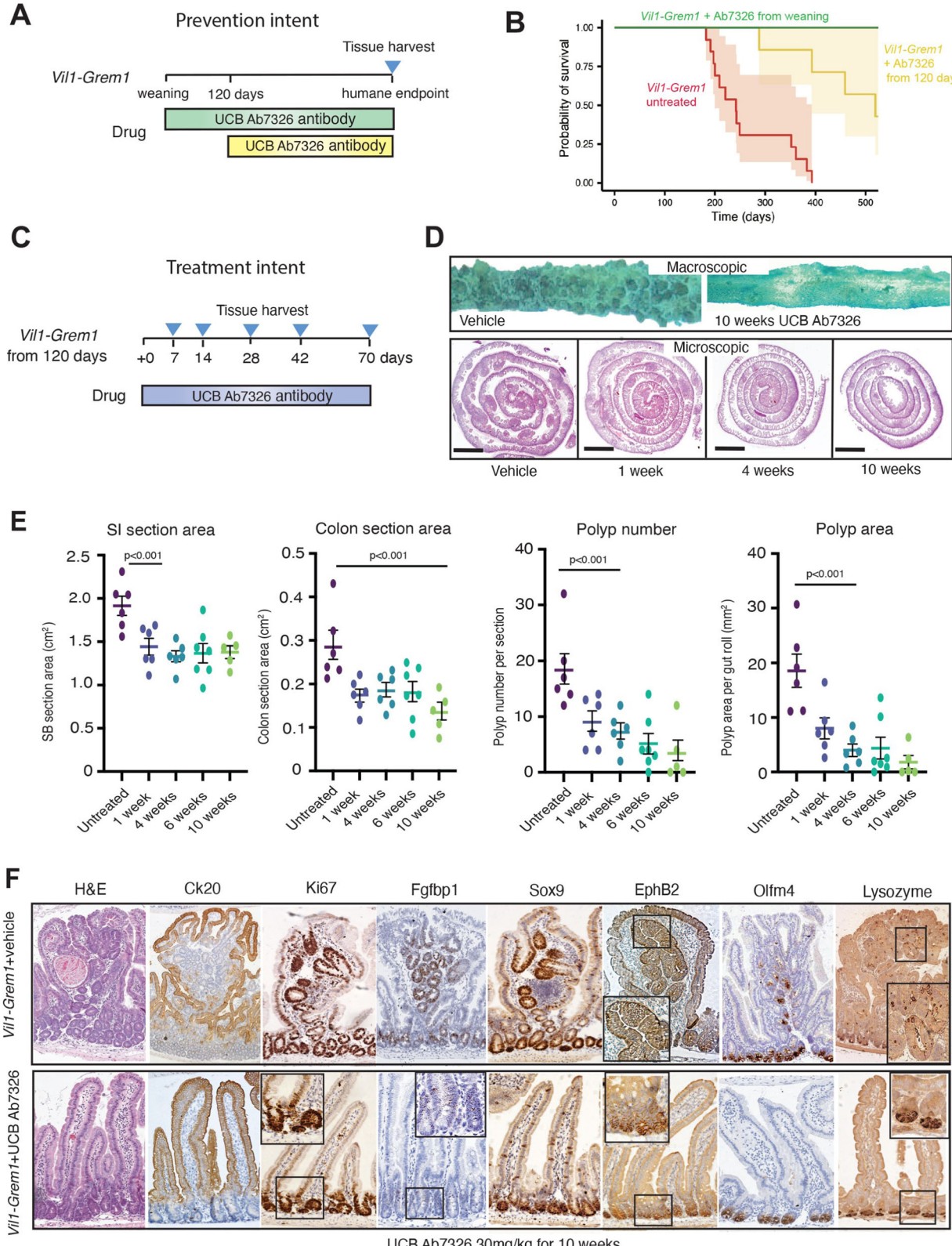

**Fig. 2 | Phenotype reversal through UCB Ab7326 treatment. A** Schematic shows treatment schedule and harvesting timepoints of *Vil1-Grem1* mice treated with UCB Ab7326. **B** Kaplan–Meier Survival Curve showing impact on survival of UCB Ab7326 initiated at weaning, (green line, $n = 11$, $P = 1.5E\text{-}06$) or from 120 days (yellow line, $n = 7$, $p = 0.00421$) in comparison to untreated animals ($n = 13$, red line). Significance calculated using log-rank tests, with the Benjamini–Hochberg correction for multiple testing. **C** Schematic shows treatment schedule and harvesting timepoints of *Vil1-Grem1* mice treated with UCB Ab7326 from the age of 120 days. **D** Representative macroscopic and histological images of proximal small intestinal sections harvested from untreated and UCB Ab7326-treated *Vil1-Grem1* mice over a range of timepoints ($n = 6$, scale bars 0.25 cm). **E** Quantification of *Vil1-Grem1* small bowel and colonic surface area, polyp number and polyp area following variable time treatment with UCB Ab7326 antibody ($n = 6$ mice per group, one way ANOVA with Dunnett post-hoc corrections, $p$ values as stated). Data were ±s.e.m. **F** Immunohistochemical analysis to show reversion of *Vil1-Grem1* ectopic crypt phenotype and restoration of normal crypt-villus staining patterns of Ki67, Sox9, EphB2, Ck20 and Lysozyme following 10 weeks UCB Ab7326 therapy ($n = 5$ mice). Scale bar 200 μm. Source data are provided as a Source Data file.

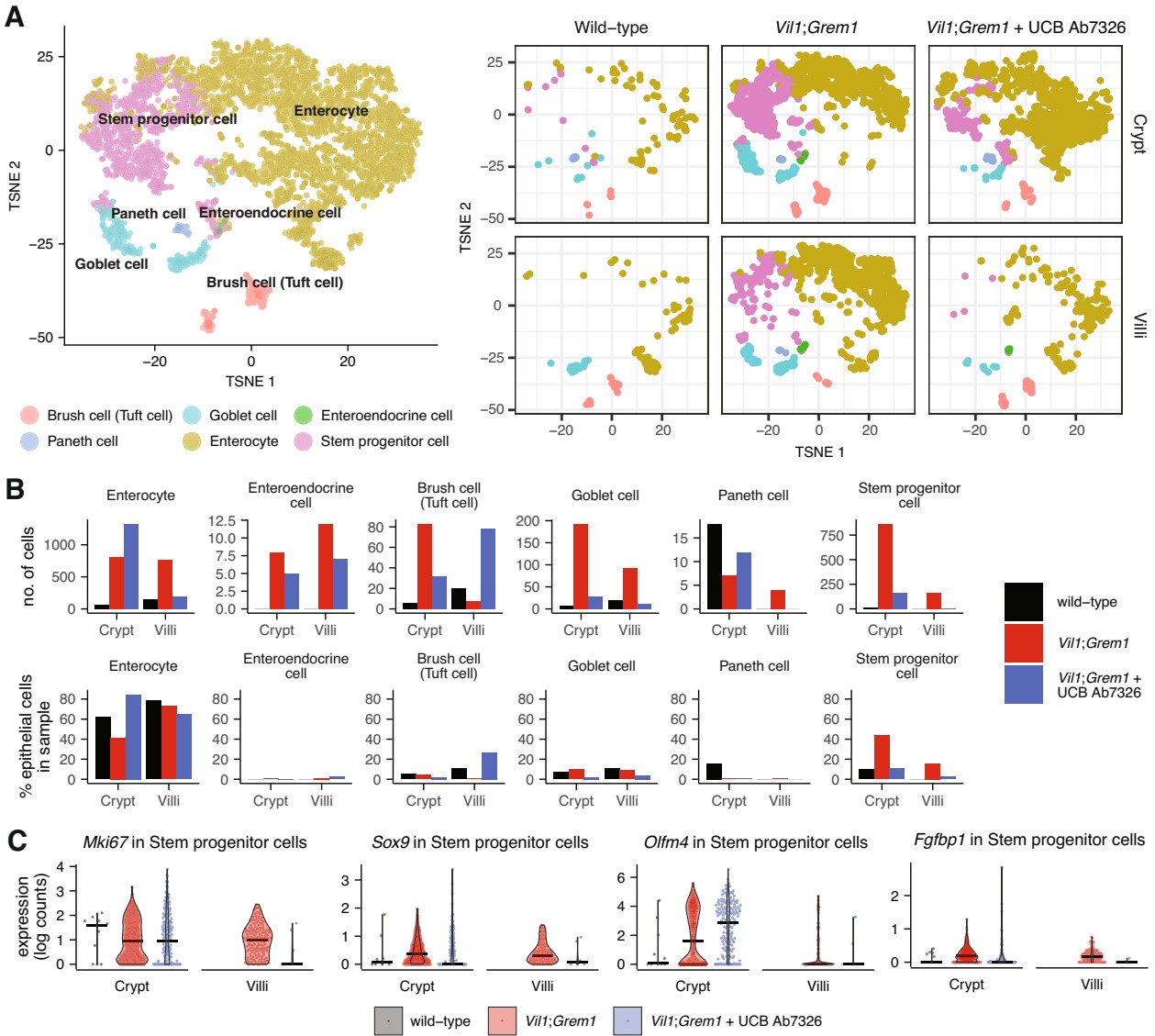

**Fig. 3 | Single-cell RNA-sequencing of proximal small bowel crypts and villi reveal an expansion of an *Lgr5*(−) intestinal stem/progenitor population that is reversed after UCB Ab7326 treatment. A** T-SNE plot of *Epcam*-expressing epithelial cells in the proximal small bowel of wild-type, *Vil1-Grem1*, and UCB Ab7326-treated *Vil1-Grem1* mice, classified by cell type, with accompanying subplots separated by mouse genotype/treatment and tissue compartment **B** The number of cells of each epithelial cell type, as well as the corresponding percentage of total epithelial cells of each cell type for each sample. The number of stem progenitor cells in either the crypt or villi/polyps show a substantial expansion of the stem progenitor cell population in both the crypt and villi of the *Vil1-Grem1* mice, which is reversed after subjecting the *Vil1-Grem1* mice to 10 weeks of treatment with UCB Ab7326. **C** The expanded stem progenitor cell population in the *Vil1;Grem1* mice is characterised by increased expression of *Mki67*, *Sox9*, *Olfm4* and *Fgfbp1*.

## Aberrant pan-epithelial Grem1 expression causes villus inter-compartmental remodelling with de-repression of stem cell-supporting fibroblast cells

In the normal intestine, Kraiczy et al., have recently shown that proximity to luminal BMP ligand source represses stem cell niche fibroblast cell functionality, whereas at the crypt base distance from ligand source and the expression of *Grem1*, relieves this repression, favouring stem cell supporting stromal cell phenotypes[8]. In light of this, and the lack of impact of autocrine *Grem1* expression on cell-intrinsic epithelial cell fate (Fig. 1E, F), we reasoned that pathological pan-epithelial expression of secreted *Grem1* in *Vil1-Grem1* animals could have an intercompartmental impact on underlying cell populations. Furthermore, the success of UCB Ab7326 treatment in phenotypic reversal provided a tool to assess the impact of partial restoration of polarised BMP signalling on different cell compartments. To investigate this, we used a variety of technologies to spatially assess

the epithelial, stromal, matrix and immune compartments in UCB Ab7326-treated and untreated *Vil1-Grem1* mice.

To assess the distribution of BMP pathway activity along the crypt-villus axis, we used chromogenic pSMAD1,5 staining. In untreated animals, aberrant epithelial Grem1 secretion not only abrogated epithelial pSMAD1,5 staining (Fig. 4A, B) but also altered the proportion, distribution and intensity of fibroblast cell expression (Fig. 4C), indicating disruption of both epithelial and stromal BMP signalling gradients. Treatment with UCB Ab7326 antibody did not eliminate epithelial autocrine Grem1 function and restore epithelial homeostatic BMP signalling gradients−marked by ongoing loss of epithelial pSMAD1,5 staining (Fig. 4A). In contrast, antibody treatment did mostly revert villus stromal BMP signalling, with re-emergence of crypt isthmus and villus fibroblast pSMAD1,5 staining (Fig. 4B).

Given the impact of epithelial *Grem1* expression on the underlying stroma, we generated a custom Hi-Plex *ISH* panel to explore the effect

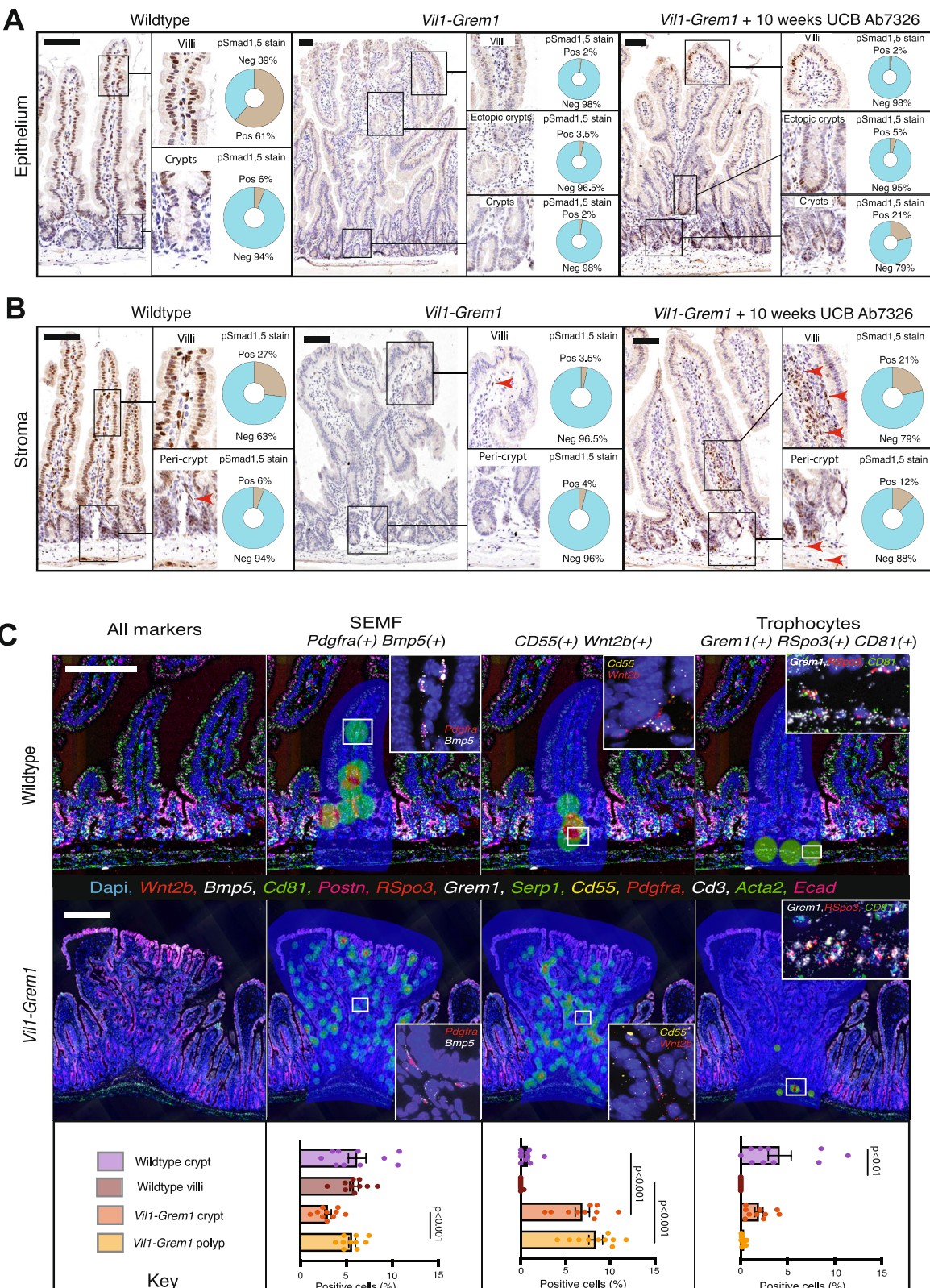

of disrupted BMP gradients on the spatial distribution of different homeostatic functional mesenchymal cell phenotypes[6,8] (Fig. S4). In wildtype animals we were able to confirm the crypt isthmus and villus distribution of SEMF's, the sub-cryptal distribution of trophocytes and the restricted peri-cryptal distribution of Cd55(+) Wnt2b(+) cells. However, in *Vil1-Grem1* animals, the crypt-restriction of Cd55(+) Wnt2b(+) cell populations was lost. Analysis across different

morphological stages of lesion development from villus enlargement through ectopic crypt emergence and established polyp development in *Vil1-Grem1* animals demonstrated de-repression of the pericrypt Cd55(+), Wnt2b(+) expressing cell population, which expanded luminally, along the length of the crypt and up into the villi, with coalescence around ectopic crypts in developing polyps (Figs. 4C, S4 and S5A−C).

**Fig. 4 | Aberrant pan-epithelial *Grem1* expression causes intercompartmental remodelling with de-repression of stem cell supporting fibroblast cells. A** p-Smad1, 5 immunohistochemistry shows physiological villus epithelial cell nuclear staining from the crypt isthmus upwards (brown stain). Pan-epithelial expression loss is seen in *Vil1-Grem1* mice, with little reversion of epithelial stain seen following 10 weeks UCB Ab7326 therapy. Automated stain quantification using digital pathology platform used to exclude stromal cell staining (QuPath) (*n* = 3 per group, >15 villi or polyps total). **B** p-Smad1,5 immunohistochemistry shows physiological stromal cell staining in the villus stromal compartment (brown stain). Loss of expression is seen in untreated *Vil1-Grem1* animals as a consequence of inter-compartmental BMP antagonism. Treatment with UCB Ab7326 recovers stromal expression levels of p-Smad1,5 in treated *Vil1-Grem1* mice. Automated stain quantification using digital pathology platform used to exclude epithelial cell staining (QuPath, *n* = 3 per group, >15 villi or polyps total). **C** Hi-plex in situ hybridization (12 markers) was used to identify and topographically map functionally distinct fibroblast subtypes (SEMF, Cd55(+) Wnt2b(+), and Trophocytes) in wildtype and *Vil1-Grem1* mouse tissue. Image analysis software was used to identify each subtype and map cell location back onto the tissue based on density distribution (coloured circles, within shaded areas - HALO, Indica Labs) (*n* = 5 mice per group, ≥10 villi, crypts or polyps total, one way ANOVA with Dunnett post-hoc corrections, *p* values as stated). Data were ±s.e.m. Scale bars 200 μm. Source data are provided as a Source Data file.

Consistent with this remodelled stromal cell landscape, we also saw significant differences in the proportion and distribution of extracellular matrix proteins in *Vil1-Grem1* animals, with an expansion of collagen1, MMP-3 and laminin expression in the polyps (Fig. S5D). Multiplex staining of key innate and adaptive immune cell populations revealed a significant influx of T-helper and T-reg cells and a reduction of T-cytotoxic cells in polyps (Fig. S5E). Previous work has shown that *Grem1*-expressing fibroblastic reticular cells in secondary lymphoid structures support dendritic cell function and provide a niche for CD4(+) T cells[16], although the functional role of *Grem1* expression in this process was not directly tested. These data provide circumstantial evidence of a possible role for homeostatic BMP signalling in maintaining normal T-cell immunity, however further study on mechanisms of immune cell regulation would be needed and are beyond the scope of this study.

Together, this work shows that disruption of homeostatic stromal BMP signalling gradients through epithelial *Grem1* expression has an impact on sub-epithelial fibroblast populations causing a de-repression of Cd55(+) Wnt2b(+) fibroblasts, which then aberrantly expand beyond the normal spatial confines of the peri-crypt region.

## Formation of the ectopic stem cell niche requires ligand-dependent Wnt signalling

Following our demonstration of BMPi-dependent remodelling of the villus stromal cell compartment, we wished to explore the signalling pathway constituents of the ectopic stem cell niche. To do this, we combined IHC and in situ hybridisation to identify morphogens expressed in ectopic crypts and their supporting stromal cells.

First, we spatially assessed the activity of the Wnt pathway in ectopic crypts at ligand, receptor and target gene level. In comparison with normal small intestinal tissue, in situ hybridisation of key intestinal wnt ligands (Figs. 5A and S6A) showed aberrant distribution and increased expression of *Wnt2b* predominantly around the ectopic crypts with some, but not all, of this ligand coming from co-stained Cd55(+) fibroblasts (Fig. 5A,E). Increased non-canonical *Wnt5a* ligand expression was seen, arising from a different stromal cell population, co-stained with periostin and situated peripherally around the polyp edge (Figs. 5A and S6B). At a receptor level, both *Fzd5* and *Lgr4* were heavily upregulated in the epithelium of *Vil1-Grem1* mice, especially in ectopic crypt cells (Fig. 5A). From a target gene perspective, both *Sox9* and *Axin2* were specifically aberrantly upregulated in proliferating ectopic crypts within the villus compartment (Figs. 1A and 5A). Following UCB Ab7326 treatment, and the restoration of crypt/villus architecture, we saw reversion of most wnt pathway activity markers to appropriate restricted expression in the crypts/isthmus.

Given this evidence for significant aberrant wnt pathway activity in ectopic crypts we undertook pharmacological and genetic manipulation of the pathway. Firstly, we used a specific Porcupine inhibitor LGK-974 to inhibit palmitoylation and secretion of wnt ligands in 120-day-old *Vil1-Grem1* and control wildtype animals (Figs. 5B–D and S6C). Both short (2 week) and long-term (10 week) (Fig. 5B) inhibition of wnt ligand secretion led to a significant reduction in polyp number, size,

ectopic crypt formation and aberrant villus cell proliferation in *Vil1-Grem1* animals (Fig. 5C, D). Porcupine inhibition did reduce abnormal villus compartment wnt activity and abrogate ectopic crypt formation, however unlike UCB Ab7326 antibody therapy, wnt ligand suppression did not restore macroscopically normal small intestinal architecture, with ongoing villus enlargement and deformity seen throughout the small bowel (Fig. 5D). To interrogate this ongoing macroscopic change, we assessed key stromal cell populations in treated mice. Following UCB Ab7326 treatment we had noted a complete re-suppression of villus Cd55(+) Wnt2b(+) cells, however this was not the case after porcupine treatment where we saw ongoing disruption of stromal cell functional architecture and continuing presence of de-repressed Cd55(+) Wnt2b(+) cells in the villus compartment (Fig. 5E).

Next, to interrogate signal transduction of ligand-dependent wnt signalling, we generated a *Rosa-CreER^T2; Lgr4-fl/fl;Vil1-Grem1* model and assessed the impact of acute loss of *Lgr4* receptor expression on ectopic crypt proliferation in 120-day-old animals with an established *Vil1-Grem1* phenotype (Fig. 6A). Following *Lgr4* receptor knockout, we saw a drop in ectopic crypt cell proliferation 3 days after recombination, implicating environmental wnt ligand-dependency in ectopic crypt maintenance (Fig. 6B). Interestingly, there was no equivalent impact on crypt basal cell turnover, implicating a reduced reliance of normal crypts on *Lgr4* receptor signal transduction (Fig. 6C). Together, this combination of genetic and pharmacological manipulation of wnt signalling demonstrates the environmental ligand-dependency of the ectopic niche in early-stage lesions, and implicates ectopic crypt epithelial cell expression of wnt receptors as a key signal transduction component.

## Advanced polyps acquire wnt disrupting mutations to bestow epithelial wnt autonomy

Consistent with previous findings[12], we saw membranous β-catenin staining and absence of *Lgr5* expression in the ectopic crypts of early lesions, but did notice nuclear β-catenin and *Lgr5* activation in a small number of large, advanced polyps in older animals. In light of the dependence of ectopic crypt formation on secreted wnt signalling in earlier stage lesions, we hypothesised that these larger polyps somatically acquire wnt disrupting epithelial mutations. Further analysis of these large lesions revealed an overlap of nuclear β-catenin positivity with activation of both *Lgr5* and *Notum* staining and concomitant down-regulation of the wnt receptor *Fzd5* in the regions of these lesions with advanced dysplasia (Fig. 6D). Targeted sequencing of dissected advanced polyps revealed epithelial wnt activating mutations in 8/11 (72%) of Lgr5(+) lesions with activating β-catenin mutations in seven lesions and a *Ptprk-Rspo3* fusion mutation in one lesion, confirmed with epithelial *Rspo3* staining on ISH (Fig. 6E). This frequency of somatic wnt disrupting mutations in large *Vil1-Grem1* polyps with advanced dysplasia is consistent with the previously noted frequency of *APC* mutations in advanced HMPS polyps in humans[12]. Together these data are consistent with a shift away from canonical wnt ligand signal transduction in advanced lesions, and reflects a switch from environmental ligand-dependency to acquired epithelial

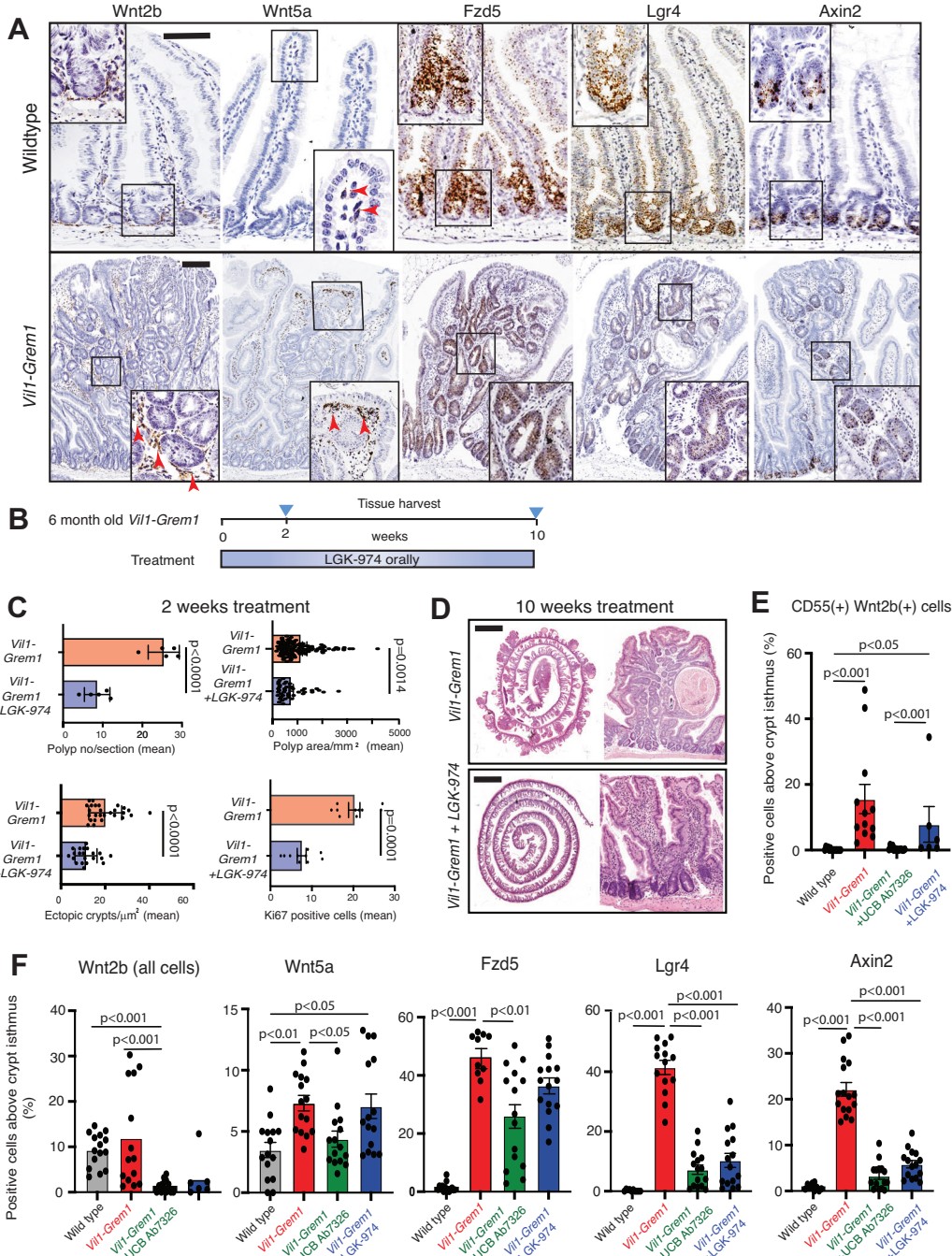

**Fig. 5 | Formation of the ectopic niche requires ligand-dependent Wnt signalling. A** Representative images of in situ hybridisation of Wnt ligands (*Wnt2b, Wnt5a*), receptors (*Fzd5, Lgr4*) and target gene (*Axin2*) in *Vil1-Grem1* versus wildtype small intestine (*n* = 5 mice). Scale bar 200 μm **B** Schematic shows treatment and harvesting timepoints of *Vil1-Grem1* animals treated with the porcupine inhibitor, LGK-974. **C** Digital pathology-based quantification of polyp number per section, polyp area, ectopic crypts and villus Ki67+ cell number per gut roll following 2 weeks of porcupine inhibition (*n* = 5 mice per group, two-tailed *t* test, *p* values as stated). Data were ±s.e.m. **D** Representative H&E images of untreated and LGK-974-treated *Vil1-Grem1* mouse small intestine (10 weeks) showing loss of polyps and ectopic crypts but ongoing villus widening/deformity in treated animals. Scale bar 0.25 cm for gut rolls **E** Quantification of Cd55(+) Wnt2b(+) cell proportion above the crypt isthmus in wildtype and *Vil1-Grem1* animals including after UCB Ab7326 and LGK-974 therapy (*n* = 5 mice, ≥15 villi or polyps total, one way ANOVA with Dunnett post-hoc corrections, *p* values as stated). Data were ±s.e.m. **F** Quantification of wnt target, receptor and target stained cell proportion above the crypt isthmus in wildtype and *Vil1-Grem1* animals including after UCB Ab7326 and LGK-974 therapy (*n* = 5 mice per group, one-way ANOVA with Dunnett post-hoc corrections, *p* values as stated). Data were ±s.e.m. Source data are provided as a Source Data file.

cell wnt autonomy as polyps progress. Notably, these advanced lesions also downregulated epithelial *Grem1* expression, through loss of the *Villin* differentiation marker in high-grade dysplasia, and acquired constitutive wnt activation through epithelial mutation rendering them insensitive to both UCB Ab7326 antibody and porcupine effect.

Together these data demonstrate reversible dependence of ectopic crypt stem cell function and proliferation on secreted, microenvironmental wnt ligand activity. Remodelled stroma acts as a key source of wnt ligand expression, through *Grem1*-induced de-repression of stem cell niche supporting fibroblasts. Although this can

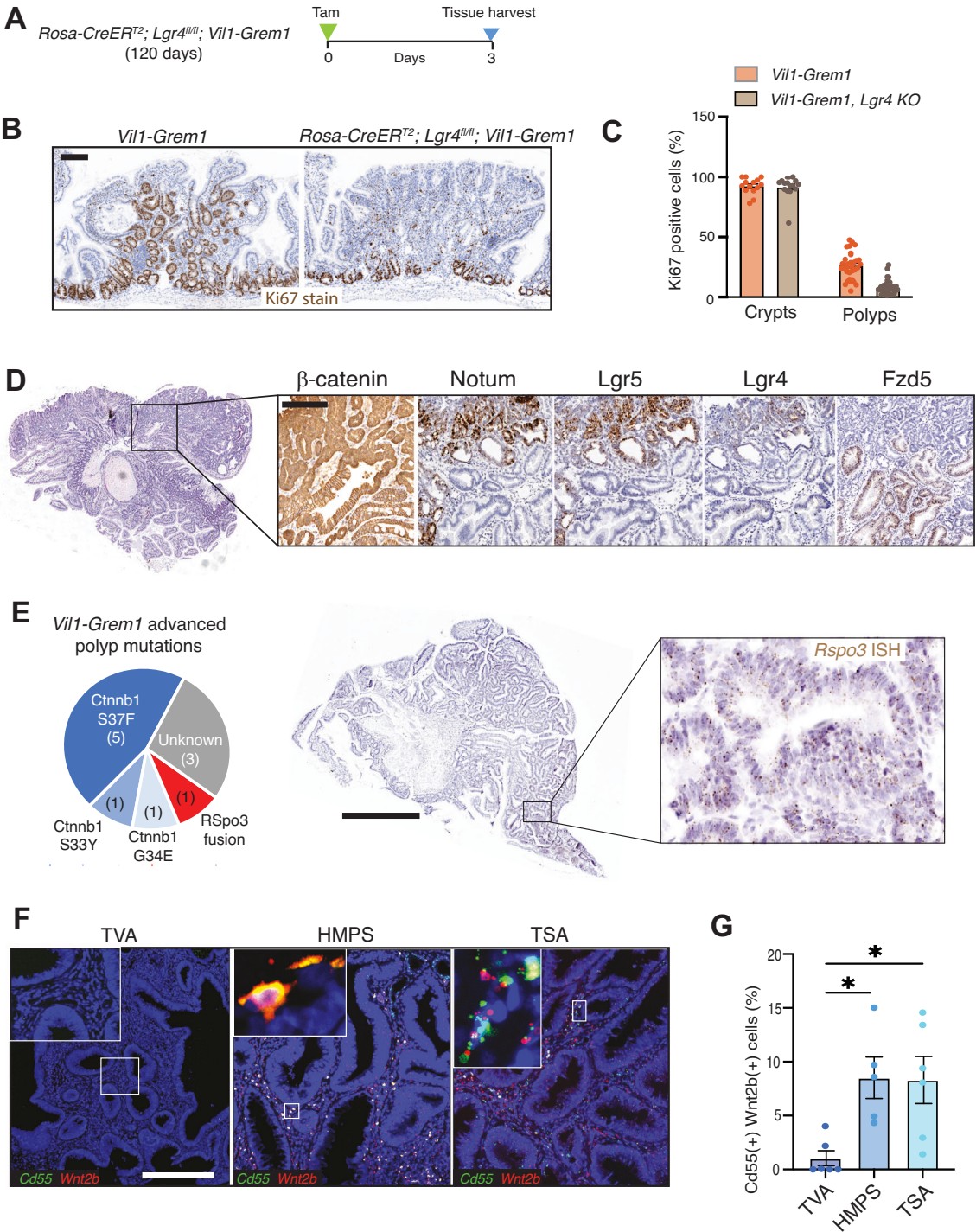

**Fig. 6 | Investigating ectopic niche and advanced polyp signalling. A** Schematic shows recombination and harvesting timepoint for acute *Lgr4* knockout in 120-day-old *Rosa-CreER^T2, Lgr4^fl/fl; Vil1-Grem1* mice. **B** Ki67 staining and **C** quantification of proliferating crypt base and ectopic crypt cells following acute *Lgr4* knockout (*n* = 15 crypts and *n* = 35 polyps total per group with *n* = 5 mice per group, t test, *P* < 0.001). Data were ±s.e.m. **D** Change in wnt target and receptor staining in advanced *Vil1-Grem1* polyps (*n* = 11 advanced polyps). **E** Pie chart shows identified mutations in *Ctnnb1*, detected by Sanger sequencing from micro-dissected advanced (*Lgr5*(+)) polyps in *Vil1-Grem1* animals (*n* = 11 advanced polyps). ISH confirms epithelial staining of *RSpo3* in a single polyp with an identified *Ptprk-Rspo3* fusion mutation. **F** Representative fluorescent co-ISH staining images, and **G** quantification of CD55(+) WNT2B(+) cell proportions in the stroma of small sporadic tubulovillous adenoma (TVA) (*n* = 6), Hereditary Mixed Polyposis Syndrome (HMPS) (*n* = 5) and sporadic traditional serrated adenoma (TSA) (*n* = 6) lesions, one way ANOVA with Dunnett post-hoc corrections, (*P* = 0.05). Data were ±s.e.m. Scale bars 200 μm. Source data are provided as a Source Data file.

be therapeutically blocked by porcupine inhibition which successfully prevents villus stem cell activity, inhibition of wnt in the ectopic niche does not lead to comparable resolution of the stromal functional architecture induced through inhibition of paracrine *Grem1*.

## Human polyp subtypes with epithelial *Grem1* expression have expanded CD55(+) Wnt2b(+) mesenchymal cell populations

Having used a disease-positioned mouse model to identify the importance of stromal remodelling in the generation of wnt-dependent ectopic stem cell niche, we turned to equivalent human samples to find evidence of the same processes. We and others have previously shown that HMPS polyps and a proportion of sporadic TSA lesions have marked ectopic crypt formation with aberrant epithelial expression of Grem1[12]. These lesions progress to an advanced polyp stage through acquired wnt disruption, including a high frequency of ligand-dependent *R-Spondin* fusions and *Rnf43* mutations in TSA's[17]. In contrast, conventional tubulovillous adenomas arise through constitutive epithelial wnt activation as a consequence of initiating *APC* mutation in crypt base columnar cells, and have only stromal expression of *Grem1* (Fig. S7). In light of the preclinical findings identifying a role for stromal remodelling and wnt ligand dependency we assessed some very small human HMPS, TSA and conventional adenoma lesions to look for evidence of de-repressed CD55(+) Wnt2b(+) fibroblast cells. We were able to identify these cells in significant numbers surrounding ectopic crypts in HMPS and TSA lesions, but could find very few within the stroma of the conventional TVA's (Fig. 6F, G). This indicates that stromal remodelling with de-repression of a CD55(+) Wnt2b(+) stromal cell population is a shared feature in mouse and human lesions characterised by ectopic crypt formation, but is not seen in Wnt ligand-independent tubulovillous adenomas where constitutive epithelial wnt activation negates the need for environmental wnt ligand supply.

## Discussion

Strictly controlled, counterbalanced secreted signalling pathways underpin the unique structure and function of the intestinal mucosa. Continual epithelial-mesenchymal crosstalk defines mucosal architecture during organ development[7] and underpins tissue homeostasis. The impact of this interaction on homeostatic epithelial cell fate is long established[3], however, the role of morphogens in determining stromal cell function is less well understood. Recently, the Shivdasani group has explored the anatomy of intestinal stromal cell heterogeneity, demonstrating a functionally layered fibroblast structure[8], dependent on cell position within mesenchymal-secreted BMP signalling gradients. In this model, proximity to a luminal surface BMP ligand source suppressed the stromal expression of stem cell niche-supporting genes, with appropriate de-repression only occurring at the crypt base through distance from the luminal BMP ligand source and the sub-crypt presence of secreted BMPi[8]. This has been further supported by the development of a model system to mechanistically interrogate the study of epithelial-fibroblast self-organisation in vitro[18]. Lin et al., demonstrate the requirement for stromal BMP responsiveness for the subsequent development of epithelial crypt-like structures, indicating a key role for stromal cell organisation in promoting epithelial cell fate polarisation and crypt base niche formation[18]. The sensitivity of the stromal cells to BMP regulation was further demonstrated in vivo by Ouahoud et al., who used deletion of stromal *Bmpr1a* receptors to alter the stromal cell secretome and initiate polyp formation[19]. Together, this work represents a shift in the epithelial-centric intestinal cell fate paradigm to a co-evolutionary model, where the interface between functionally distinct epithelial and stromal cell populations is established through shared exposure to paracrine signalling concentration gradients. Thus, crypt-villus compartmentalisation is dependent on bidirectional crosstalk between epithelial and stromal cells, with each

cell population influencing and regulating the fate and function of each other. This cell compartment interdependence enables the tissue to be exquisitely sensitive and responsive to local change, which is required for a rapid and adaptive physiological response to damage[20], but also renders the mucosa susceptible to pathological dysregulation, especially following mutation-induced disruption of the key regulatory pathways.

Here, we used a disease-positioned model of a rare human polyposis syndrome to mechanistically explore the impact of pathological disruption of polarised BMP gradients and provide insight into the co-dependence of epithelial-mesenchymal fate and function in the gut. Using specific pathway inhibitors, we were able to triangulate the spatio-temporal consequences of cumulative signalling pathway disruption and define a pathological sequence of events. We demonstrate that aberrant pan-epithelial *Grem1* expression acts intercompartmentally to disrupt BMP signalling gradients in both epithelium and underlying stromal cell populations. Resultant remodelling of the stromal cell compartment results in the de-repression and expansion of Cd55(+) Wnt2b(+) fibroblasts which coalesce and surround developing ectopic crypts and are capable of supporting non-conventional stem cell function. Ultimately this leads to the formation of a wnt ligand-dependent ectopic stem cell niche outside of the confines of the crypt base. Although abrogation of wnt ligands through porcupine inhibition does prevent ectopic crypt formation and significantly suppresses advanced polyp formation, it is only attenuation of aberrant paracrine BMP antagonism that is capable of reverting fibroblast remodelling and allowing re-imposition of appropriate homeostatic stromal cell functional segregation. The dependence of this permissive ectopic stem cell niche on paracrine signalling renders the phenotype therapeutically reversible in advance of acquisition of epithelial somatic mutation. This work demonstrates that aberrant epithelial *Grem1* expression disrupts homeostatic BMP crosstalk between the epithelium and the underlying mesenchyme, and transforms the mucosal stromal cell architecture into a microenvironment capable of sustaining an ectopic stem/progenitor cell population. Together, this illustrates the pathological consequence of disruption of homeostatic morphogen gradients in a human disease-relevant setting, and supports an emerging co-evolutionary model of intestinal cell compartmentalisation based on bidirectional regulation of epithelial-mesenchymal cell fate and function (Fig. 7).

In established cancers, *GREM1* expression is frequently upregulated in the desmoplastic stroma of a number of solid tumours[21]. In sporadic colorectal cancer, high stromal *GREM1* expression is clearly associated with aggressive mesenchymal subtypes and poor prognosis[12]. As we have demonstrated a role for *GREM1* in remodelling the stromal microenvironment to generate an ectopic stem cell niche in a disease-positioned model of a hereditary intestinal polyposis, it would be interesting to explore whether this mechanism persists into established sporadic cancers. This represents a potential therapeutic opportunity given the increasing interest in the role of the tumour microenvironment in supporting both *LGR5*(+) and (−) cancer stem cell populations[22,23]. Other recent publications have also implicated a role for *Grem1* in regulating epithelial-mesenchymal transition[24] and lineage plasticity through direct activation of the fibroblast growth factor receptor 1[25] in pancreatic cancer and prostate cancer, respectively. The latter effect is intriguing given the demonstrated link between fibroblast growth factor signalling and a possible *Lgr5 (-)* upper crypt stem cell population in the intestine[5,6], as we see clear expansion of *Fgfbp1* expressing cells into villus ectopic crypts in our model (Fig. 1A). Together this provides accumulating evidence for a possible pleiotrophic role for *GREM1* inhibition in neoplasia and requires further detailed study.

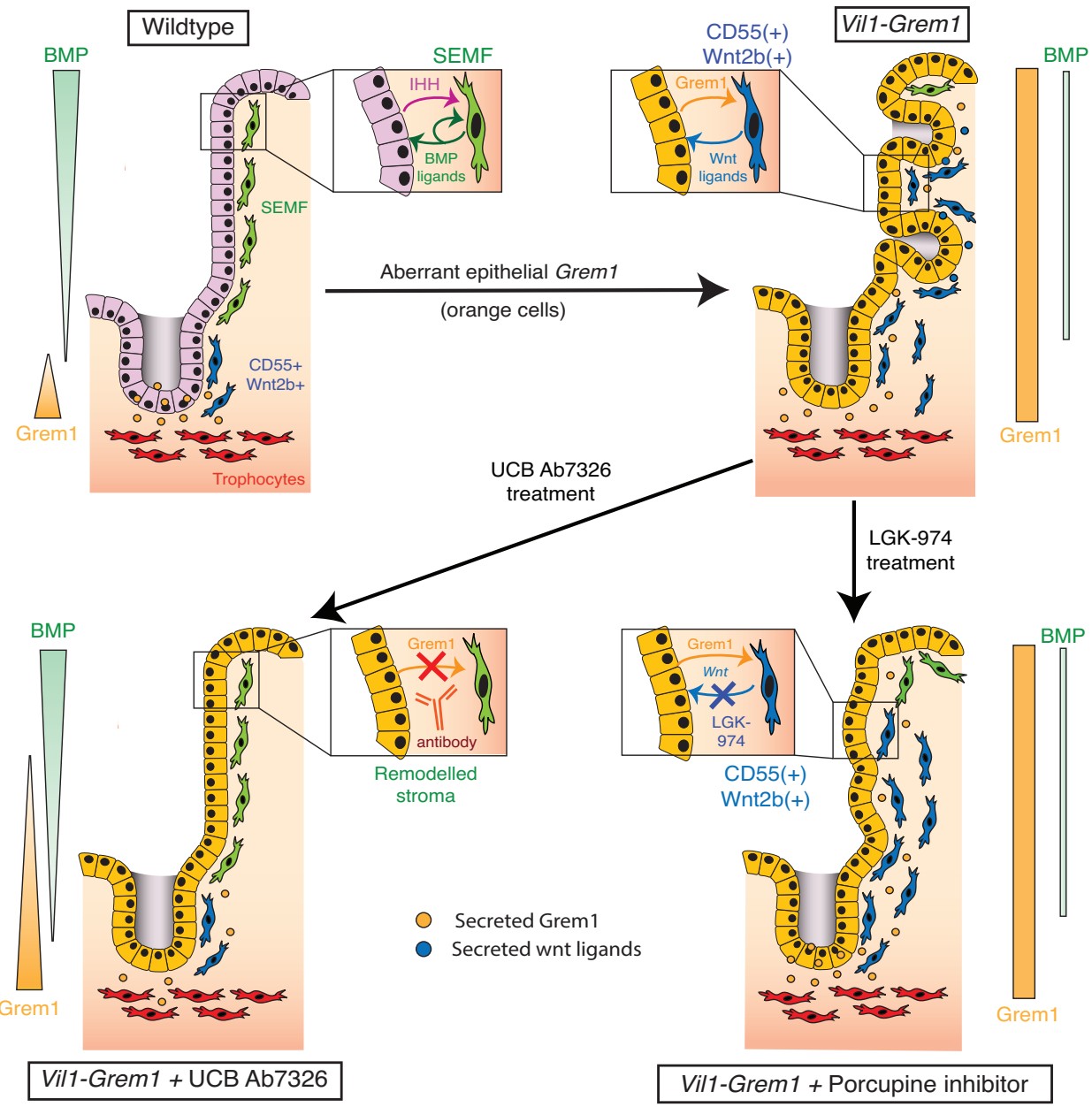

**Fig. 7 | Human polyp stromal remodelling and model summary.** Model summary depicting the disruption of BMP signalling gradients leading to a remodelling of the stromal microenvironment and establishment of wnt ligand-dependent ectopic crypt stem cell niches.

What is clear from this work is that the remarkable phenotypic reversion and long-term survival of *Vil1-Grem1* mice commenced early on UCB Ab7326 antibody therapy, providing hope that *Grem1* inhibition could offer a preventative therapeutic strategy for HMPS patients. This hereditary polyposis carries a high lifelong cancer risk and prophylactic treatment could potentially reduce the need for regular intrusive surveillance colonoscopies or prophylactic colectomy in this rare patient population.

## Limitations of this study

The model presents a framework for understanding ectopic crypt initiation, although the precise mechanism remains uncertain. It is unclear whether epithelial invagination results from physical buckling due to an expanded villus cell compartment (as proposed by Shyer et al.) or as a response to an expanding underlying stromal Wnt source. Nonetheless, this concept aligns with the view that ectopic crypt formation arises from Bmp signalling disruption, affecting the interdependent and co-evolving epithelial and stromal cell compartments.

## Methods

### Resource availability

Further information and requests for resources and reagents should be directed to and will be fulfilled by the lead contact Hayley Belnoue-Davis (hayley.davis@well.ox.ac.uk).

### Experimental model and subject details

**Animals.** All procedures were carried out in accordance with Home Office UK regulations and the Animals (Scientific Procedures) Act 1986 (under project licences P0B63BC4D, PP0226443, 7009112 and PP3908577) and by adhering to the ARRIVE guidelines with approval from the local Animal Welfare Ethical Review boards of University of

Oxford (AWERB Clin Med) or the University of Glasgow. All mice are housed in individually ventilated cages at the animal unit either at the Functional Genetics Facility (Centre for Human Genetics, University of Oxford) or The Beatson Institute (Glasgow). All mice were housed in a specific-pathogen-free (SPF) facility, with unrestricted access to food and water, and were not involved in any previous procedures. All strains used in this study were maintained on a C57BL/6J background for ≥6 generations. Procedures were carried out on mice of at least 6 weeks of age, both male and female. Following timed mating, embryos were collected at E16.5 and post-natal pups were collected at P0.5 and P10. DNA was extracted from the tails to use for genotyping.

**Generation of *Rosa26^Grem1* mouse.** To generate *Rosa26^Grem1* mice, a Grem1 complementary DNA cassette was cloned into the integrase-mediated cassette exchange vector (CB93) and transfected into RS-PhiC31 ES cells. Recombinant clones were obtained which harbour the Grem1 complementary DNA transgene positioned within the Rosa26 locus, allowing for Cre-dependent activation of transgene expression. Recombinant clones were injected into blastocysts, and chimeras were generated (Centre for Human Genetics Trangenics Core). Chimeras were crossed with wildtype C57BL/6J mice to obtain F1 heterozygotes.

**Human subjects.** Samples were collected from the Oxford GI biobank OCHRe: 22/A158 and 22/A158b. All samples were subject to expert histopathological review.

**Treatment of animals.** The mouse alleles used in this study in various combinations are as follows: *Vil1-Grem1, Atoh1CreER^T2, Villin-CreER^T2*[26], *RosaCreER^T2* [27], *Sox9CreER^T2* [28] *Rosa26^Grem1*, *Rosa26^tdTom*[29] *Rosa26^YFP*[30], *Lgr4^fl/fl*[31]. All experimental mice were from a C57/BL6 background, backcrossed for at least 6 generations, and were housed in specific-pathogen-free cages. For inducible models, recombination was achieved using the free base tamoxifen (Sigma-Aldrich, St. Louis, MO) dissolved in ethanol/oil (1:9). For lineage tracing experiments (*Atoh1CreER^T2;Rosa26^tdTom* and *Atoh1CreER^T2;Rosa26^Grem1*), recombination was induced by a single dose of 3 mg of tamoxifen. For anti-Grem1 therapy mice received weekly 30 mg/kg subcutaneous administration of UCB Ab7326 (UCB Pharma). To generate Kaplan-Meier data, mice were sacrificed when they reached humane-end points (exhibited anaemia, hunching and inactivity). For WNT blocking LGK974 (MedKoo, 205851) was used. LGK974 was administered at a concentration of 5 mg/kg, in 0.5% Methylcellulose twice daily by oral gavage.

**Formalin-fixed paraffin-embedded processing.** Gut preparations were washed in PBS, fixed overnight in 10% neutral buffered formalin and then transferred to 70% ethanol before processing for embedding. Formalin-fixed gut sections were rolled into Swiss Rolls, pinned and placed in a histology cassette. Specimens were processed using a Histomaster machine (Bavimed). Processed samples were embedded in paraffin wax using a paraffin embedding station (EG1150H, Leica).

**In situ hybridization.** For in situ hybridization (ISH) of both human and mouse FFPE samples, 4 µm formalin-fixed, paraffin-embedded tissue sections were used. The sections were baked at 60 °C for 1 h before dewaxing in xylene and ethanol. Chromogenic ISH was performed using the RNAscope® 2.5 HD -BROWN and Fluorescent ISH was then performed using the RNAscope Fluorescent Multiplex Reagent Kit (Bio-techne) in accordance with the supplier's guidelines. All probes were purchased from ACD (Bio-techne),and included: Mm-Grem1 (ACD, Cat# 314741), Mm-Wnt5a (ACD, Cat# 316791); Mm-Rspo2 (ACD, Cat# 402001); Mm-Wnt2b (ACD, Cat# 405031); Mm-Notum (ACD, Cat# 428981); Mm-Lgr4 (ACD, Cat# 318321); Mm-Fzd5 (ACD, Cat# 404911); Mm-Dll1 (ACD, Cat# 425071); Mm-Jag1 (ACD, Cat# 412831); Mm-Axin2 (ACD, Cat# 400331); Mm-Fgfbp1 (ACD, Cat# 508831); Mm-

Sox9 (ACD, Cat# 401051-C2); Hs-CD55 (ACD, Cat# 426551); Hs-Wnt2b (ACD, Cat# 453361-C2).

**Immunohistochemistry.** Sections were de-paraffinized in xylene and rehydrated through graded alcohols to water. Antigen retrieval was achieved by pressure cooking in 10 mmol/L citrate buffer (pH 6.0) for 5 min. Endogenous peroxidase activity was blocked by incubating in 3% hydrogen peroxidase (in methanol) for 20 min. Next, sections were blocked with 1.5% serum for 30 min, after which they were incubated with primary antibodies for 1 h. The sections were then incubated with appropriate secondary antibodies for 30 min at room temperature. For chromogenic visualization, sections were incubated with ABC (Vector labs) for 30 min and stained using DAB solution (VectorLabs), after which they were counterstained with hematoxylin, dehydrated and mounted. For Lysozyme/Alcian blue staining, directly following DAB development, slides were incubated in 1% Alcian Blue with 3% acetic acid (Sigma) for 30 min, followed by counterstaining with 0.1% Nuclear Fast Red solution (Sigma) for 5 min, and then dehydrated and mounted. All antibodies used in this work are as follows: Anti-Ki-67 (D3B5) Rabbit mAb (1:200, Cell Signalling Technology, Cat#CS12202S); Anti-lysozyme Rabbit pAb (1:1000, DAKO, Cat#EC3.2.1.17); Anti-mCherry (TdTomato) mouse mAb (1:500, Novus Bio, Cat#NBP1-96752); Anti-SOX9 Rabbit pAb (1:1000, Sigma Aldrich, Cat#AB5535); Human/Mouse EphB2 Antibody (1:125, R&D, Cat#AF467; Phospho-SMAD1/5 (1:200, Ser463/465) Rabbit mAb (Cell Signalling Technology, Cat#41D10); Anti-β-Catenin Clone 14 (1:50, BD Biosciences, Cat#610154); Olfm4 (D6Y5A) XP® Rabbit mAb (1:200, Cell Signalling Technology, Cat#39141), anti-GFP/YFP Rabbit polyclonal Ab (1:1000, ThermoFisher Scientific, Cat#A6455), anti-Cytokeratin 20 Rabbit polyclonal Ab (1:500, Abcam, ab118574).

**Multiplex immunofluorescence.** Multiplex immunofluorescence (MPIF) staining was performed on FFPE sections of thickness 4-µm using the OPAL protocol (Akoya Biosciences, Marlborough, MA) on the Leica BOND RXm autostainer (Leica Microsystems, Wetzlar, Germany). Six consecutive staining cycles were performed using the following primary antibody-Opal fluorophore pairs:

**Immune panel.** (1) Ly6G (1:300, 551459; BD Pharmingen)−Opal 540; (2) CD4 (1:500, ab183685; Abcam)−Opal 520; (3) CD8 (1:800, 98941; Cell Signaling)−Opal 570; (4) CD68 (1:1200, ab125212; Abcam)−Opal 620; (5) FoxP3 (1:400, 126553; Cell Signaling)−Opal 650; and (6) E-cadherin (1:500, 3195; Cell Signaling)−Opal 690.

**Matrix panel:.** (1) Laminin (1:400, ab11575; Abcam)-Opal 540; (2) Tenascin-C(1:600, ab108930; Abcam)-Opal 520; (3) Fibronectin (1:1000, F3648; Sigma-Aldrich)-Opal 570; (4) Osteopontin (1:750, ab218237; Abcam)-Opal 620; MMP3 (1:100,ab52915; Abcam)-Opal 650; (5) Collagen I (1:400, 72026; Cell Signaling)-Opal 690.

Tissue sections were incubated for 1 h in primary antibodies and detected using the BOND Polymer Refine Detection System (DS9800; Leica Biosystems, Buffalo Grove, IL) in accordance with the manufacturer's instructions, substituting DAB for the Opal fluorophores, with a 10-min incubation time and withholding the hematoxylin step. Antigen retrieval at 100 °C for 20 min, in accordance with standard Leica protocol, with Epitope Retrieval Solution one or two was performed prior to each primary antibody being applied. Sections were then incubated for 10 min with spectral DAPI (FP1490, Akoya Biosciences) and the slides mounted with VECTASHIELD Vibrance Antifade Mounting Medium (H-1700-10; Vector Laboratories). Whole-slide scans and multispectral images (MSI) were obtained on the Akoya Biosciences Vectra Polaris. Batch analysis of the MSIs from each case was performed with the inForm 2.4.8 software provided. Finally, batched analyzed MSIs were fused in HALO (Indica Labs) to produce a spectrally unmixed reconstructed whole-tissue image. Cell density

analysis was subsequently performed for each cell phenotype across the three MPIF panels using HALO.

**HiPlex ISH.** HiPlex ISH staining was performed on FFPE sections of thickness 4-µm and was completed as an outsourced to BioTechne who used the RNAscope™ HiPlex12 Reagent Kit (488, 550, 650, 750) v2 Standard Assay. This panel probed for the following mouse RNA transcripts: Mm-Wnt2b-T6 (ACD, Cat#405031-T6); Mm-Bmp5-T7 (ACD, Cat#401241-T7); Mm-Cd81-T5 (ACD, Cat#556971-T5); Mm-Postn-T8 (ACD, Cat#418581-T8); Mm-Rspo3-T2 (ACD, Cat#402011-T2); Mm-Grem1-T3 (ACD, Cat#314741-T3); Mm-Serpine1-T1 (ACD, Cat#402501-T1); Mm-Cd55-T4 (ACD, Cat#421251-T4); Mm-Pdgfra-T10 (ACD, Cat#480661-T10); Mm-Cd3e-T11 (ACD, Cat#314721-T11); Mm-Acta2-T9 (ACD, Cat#319531-T9); Mm-Cdh1-T12 (ACD, Cat#408651-T12).Whole-slide scans and multispectral images (MSI) were obtained on the Akoya Biosciences Vectra Polaris.

**Image analysis.** IHC images were analyzed as follows: Positive cells were quantified using QuPath digital pathology software (v0.2.3, (Bankhead et al., 2017)), downloaded from https://QuPath.github.io/. Firstly, annotations of tissue areas were created for each sample with areas of folded tissue excluded to eliminate false positive signals. Cells were identified within QuPath using a custom algorithm established via stain separation using color reconstruction. Positive cell detection analysis was run to identify DAB-positive cells and results reported as percentage of positive cells. Each annotation was manually verified for correct signal identification. For analysis of multiplex IHC, HiPlex ISH and dual ISH, HALO image analysis software (Indica Labs) was used to identify cell phenotypes, cell density analysis and mapping of cell phenotypes onto images. For 3D reconstruction of lineage-traced crypts in *Sox9-CreER^{T2}; Rosa26^{YFP}; Vil1-Grem1* polyps, images were aligned using HeteroGenius MIM with 3D Pathology AddOn (Hetero-Genius, Leeds, UK).

**Statistical analysis.** All statistical calculations were performed using GraphPad Prism 10. Unpaired two-tailed t tests were used to determine statistical significance between two groups. One-way or two-way analysis of variance (ANOVA) was used to assess statistical significance between three or more groups. For precise details on statistical analyses (including post-hoc tests), size of n, tests utilised, and significance definition ($p$-values) see corresponding figure legend and results. For individual value plots, data are displayed as mean ± standard error of the mean (s.e.m.). $P$ values < 0.05 were considered significant.

**Single cell RNA-sequencing.** Changes in the cellular composition of intestinal crypts, villi, and polyps from wildtype, Grem1-overexpressing (*Vil1-Grem1*) and anti-Grem1 antibody-treated (UCB Ab7326 mIgG, UCB Pharma, administered at 30 mg/kg subcutaneously weekly from the age of 6 weeks old for 10 weeks) *Vil1-Grem1* mice were examined by single-cell RNA sequencing.

To collect the samples, the proximal small bowel was dissected out and opened longitudinally. The muscle layer was scraped off mechanically with a coverslip and discarded. The villus compartment and crypt compartment were separated mechanically and digested using the MACS human umbilical cord dissociation kit (Miltenyi, 130-105-737) following the manufacturer's instructions, with an enzymatic digestion time of 1 h15 m. To minimise the digestion time the dissection process was undertaken macroscopically using whole intestinal tissue and we are unable to completely exclude small amounts of attached normal crypt contamination in the villus fraction. This is a limitation of the experiment. Three mice per group were pooled for each sample. Cells were stained with propidium iodide (1:500,

Invitrogen) and live cells were sorted using the BD FACSAria Fusion Cell Sorter.

Nucleic acids were directly processed on intact dissociated cells and libraries were prepared using the Chromium Single Cell 3' Reagent v3 chemistry kit (10X Genomics) on a Chromium Instrument (10x Genomics). Library quality was assessed using a TapeStation (Agilent Technologies). Prepared libraries were sequenced on a NovaSeq 6000 instrument (Illumina) at the Oxford Genomics Centre.

Cell demultiplexing and sequence alignment were performed with 10X Genomics Cell Ranger (v7.1.0) using the mm10-2020-A reference transcriptome. Counts due to ambient RNA molecules and random barcode swapping from the (raw) UMI-based scRNA-seq gene-by-cell count matrices were removed and empty droplets were filtered out using CellBender (v0.3.0). The processed single cell RNA-sequencing data were analysed in the R statistical environment using the scran (v1.32.0) and scater (v1.32.0) packages. Doublets were identified and excluded from the dataset using the scDblFinder (v1.18.0) package. Cells with mitochondrial gene expression percentages ≥15%, detected genes ≤200, and total UMI counts ≤1000 were excluded from the dataset. The data from each sample were normalized, merged, MNN-normalized, clustered using the Louvain method, and visualized by UMAP and T-SNE dimensionality reduction. Clusters of cells that showed distinct *Epcam* expression were selected from the dataset as epithelial cell clusters, and subjected to an initial semi-supervised cell type assignment against known mouse intestinal epithelial marker genes from the CellMarker 2.0 database using the algorithm in the scSorter (v0.0.2) package. Cell type annotations were further subjected to manual curation, validation, and confirmation using a range of mouse intestinal epithelial marker genes prior to further downstream data visualization and analysis.

### Reporting summary
Further information on research design is available in the Nature Portfolio Reporting Summary linked to this article.

## Data availability
The single-cell RNA-sequencing data have been made available in the BioStudies database (http://www.ebi.ac.uk/biostudies) under accession number E-MTAB-14360. Large raw image/micrograph files that support the findings in this study are not provided with the source data and are available from the corresponding authors upon request. Source data are provided with this paper.

## Materials availability
Mouse lines generated in this study are available with an MTA.

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

## Acknowledgements

The authors thank patients visiting the John Radcliffe Hospital, especially those with HMPS, for their generous tissue donation. We would also like to thank the Oxford TGLU GI Illness Biobank team for their help collecting this tissue. The authors also thank Derek Magee from HeteroGenius for digital reconstruction of lineage tracing ribbons in polyps. The authors are grateful to the Core Research Services at the CRUK Scotland Institute: Biological Services Unit and Histology services for technical support and to Catherine Winchester for critical reading of the manuscript. S.J.L. was supported by CRUK Program Grant (DRCNPG-Jun22\100002) and a Wellcome Trust (Senior Clinical Research Fellowship (206314/Z/17/Z). E.J.M. was supported by the Lee Placito Research Fellowship (University of Oxford). G.N.V. was supported by an Oxford-BMS Fellowship. J.E.E. was supported by the National Institute for Health Research (NIHR) Oxford Biomedical Research Centre. Experimental work was also supported by Rosetrees Trust and Stoneygate Trust research grant (M493) and a UCB Pharma research grant. Animal costs were supported by an International Accelerator Award, ACRCelerate, jointly funded by Cancer Research UK (A26825 and A28223), FC AECC (GEACC18004TAB) and AIRC (22795), and the MRC Mouse Genetics Network. Core funding to the Wellcome Centre for Human Genetics was provided by the Wellcome Trust (090532/Z/09/Z). This research was funded in part by the Wellcome Trust. For the purpose of Open Access, the author has applied a CC BY public copyright licence to any Author Accepted Manuscript version arising from this submission. The views expressed are those of the author/s and not necessarily those of the NHS, the NIHR or the Department of Health.

## Author contributions

E.J.M., H.B.D., I.T. and S.J.L. conceived and designed the project. Funding obtained by S.J.L. and E.J.M. Experiments were conducted by H.B.D., E.J.M., N.G., A.L., M.L., S.B., E.G.V., S.O., N.N., M.H., A.S.N. Bioinformatic analysis carried out by G.N.V. Pathology support L.M.W. Tissue, materials, and data provision K.S.M., L.K.J., R.R., S.I., J.E.E., L.M.W., N.D., T.P., G.D. Conceptual input and data interpretation G.D., N.D., O.S., I.T. Manuscript written by E.J.M., G.N.V. and S.J.L.

## Competing interests

The named authors declare the following competing interests; S.J.L. has received grant income from UCB Pharma; G.D. was a UCB employee at the time the research was conducted; N.D. is an employee of UCB Pharma, UK and owns shares in UCB Pharma and Vertex Pharmaceuticals. All other authors declare no competing interests.
