## [Transparent Peer Review file · Nature Communications]

Epithelial GREMLIN1 disrupts intestinal epithelial-mesenchymal crosstalk to induce a wnt-dependent ectopic stem cell niche through stromal remodelling

Corresponding Author: Dr Hayley Belnoue-Davis

Version 0:

Reviewer comments:

Reviewer #1

(Remarks to the Author)

Leedham lab has reported a role of Gremlin1 in stromal remodeling. They developed a therapeutic strategy using a Gremlin1 antibody and a Porcupine inhibitor that prevented ectopic crypt formation in the hereditary mixed polyposis syndrome (HMPS) mouse model. In addition, the authors found that mesenchymal gene expression patterns were altered by epithelial Gremlin1 production, which in turn supported de novo stem cell niche environments and ectopic crypt formation. These results are interesting and appropriate for publication in Nature Communications. There are some issues that need to be addressed before its publication.

1. While they provide substantial data, some are not directly relevant to the main story. Specifically, the section on Notch signaling (ATOH1-CreER tracing and DBZ (Notch inhibitor) treatment) seems unnecessary (it is not included in the model summary of Figure 6c) and complicates the story. I suggest they either remove these parts or provide a more coherent explanation of why the Notch pathway is important in stromal remodeling.
2. The authors provided almost no mechanism of how epithelial Gremlin1 remodels the stromal niche environments. The epithelial-mesenchymal interactions illustrated in Figure 6c are interesting but somewhat hypothetical, as no biological data supports this. It would be beneficial if they could provide some functional data, such as in vitro remodeling of stroma by Gremlin1. They discussed FGFR activation by Gremlin1, but I would like to see if this is validated in their model.
3. The Gremlin1 antibody treatment experiment elegantly demonstrated its therapeutic potential for HMPS mouse model. However, the authors treated the mice from 1-week old onwards, which only proved its preventative effect on tumorigenesis. I wonder if they could demonstrate whether polyps could be eliminated when treatment is started from 1-month old mice. Therapeutic data would significantly enhance the value of the antibody treatment strategy.

Reviewer #2

(Remarks to the Author)

In this manuscript Mulholland and Belnoue-Davis and colleagues describe the impact of constitutive pan-epithelial Gremlin1 over-expression on intestinal homeostasis. In particular, they describe the formation of ectopic epithelial crypts, reminiscent of HMPS-phenotypes, and the role of stromal cells in the establishment of this Gremlin1-induced phenotype. It is an extension of previous work from the same lab (<https://doi.org/10.1038/nm.3750>) that already describes the formation of ectopic crypts in this mouse model (Vil1-Grem1 mice). Here, the authors add data regarding the potential underlying mechanisms, rooted in the pharmacological reversion of the phenotype using a Gremlin1-antibody (UCB Ab7326). This finding is interesting and adds potential translational value for HMPS patients. However, the manuscript remains rather descriptive and superficial with important controls missing, as outlined below in the point-by-point.

Major comments:

- The authors do not analyze the fate of cells that depend on Bmp signaling – in particular enterocytes and enteroendocrine cells (as shown in publications by Beumer and Clevers: doi: 10.1016/j.celrep.2022.110438, doi: 10.1016/j.cell.2020.08.005) in both their Vil1-Grem1 mouse model, as well as upon Grem1 antibody (UCB Ab7326) treatment. This is a major omission and requires critical further analysis. Along similar lines, in the text corresponding to Figure 11 the authors claim: "... lead to

disrupted secretory cell fate". What exactly do the authors mean by this? They did not analyze secretory markers and they did not perform extensive expression profiling (e.g., single-cell RNA sequencing). As mentioned, Beumer and Clevers showed that Bmp signaling may affect the cell fate of goblet cells (e.g., the production of antimicrobial peptides). Therefore, as Gremlin1 inhibits Bmp signaling is goblet cell fate disrupted in Vil1-Grem1 mice?

- Another major aspect that is fully ignored by the authors is the timing of Grem1 expression in their Vil1-Grem1 mouse model. Given the nature of the mouse model, the expression of Grem1 is not inducible. As Vil1 starts to be expressed during embryonic development it is highly likely that already the development of the embryonic gut is altered and proper intestinal niches are not established to begin with, which severely affects the interpretation of the presented phenotypes. What if e.g., Grem1-induced ectopic crypts retained embryonic features due to a corrupted development (in Vil1-Grem1 animals)? The authors should investigate such issues using proper time-course analysis starting already during embryonic gut development.

- Their choice of model system leads to additional interpretative issues when the authors attempt to distinguish between the effects of Grem1 overexpression from individual secretory cells vs. the entire epithelium. Their comparison between Atoh1-CreERT2;R26-LSL-Grem1 and Atoh1-CreERT2;R26-tdTom;Vil1-Grem1 is not only a comparison between expression of Grem1 from individual secretory cells vs. pan-epithelial expression, but more importantly a comparison between acute overexpression of Grem1 in the adult intestinal context vs. constitutive Grem1 overexpression for the entire developmental history (including embryonic development) of the intestine. As such, their interpretation of the observed differences in phenotype are misleading. It would be more appropriate to compare e.g., Vil1CreERT2;R26-LSL-Grem1 (pan-epithelial) with Atoh1CreERT2;R26-LSL-Grem1 (secretory specific) mice to support the authors claims.

- Why did the authors choose Sox9 to trace the origin of the ectopic crypts and why was only partial recombination induced to trace this lineage? This can lead to incorrect observations/statements as some relevant Sox9 progenitors might not be labeled. The essential control condition however is missing – tracing Sox9 for the same timeframes in the wildtype/homeostatic intestine. The authors suggest that Sox9+(Lgr5-) cells in the Vil1-Grem1 mice have stem cell potential in the ectopic crypts. It would be nice if they could show this e.g., by isolating these cells and test their ability to form epithelial organoids (as has previously been shown for Lgr5+ cells isolated from the homeostatic intestine).

- As stated by the authors, a correct balance of Wnt- and Bmp gradients are integral to the establishment of the intestinal stem cell and differentiation niches. As such, it is very surprising that treatment with Grem1 antibody (UCB Ab7326) or Porcupine inhibition (LGK-974) result only in a reversal of the aberrant phenotypes in Vil1-Grem1 animals and do not affect homeostatic niches and as a result mouse survival at the same time. Here, again important control experiments are missing. The authors should test these treatment modalities in wildtype mice to judge the impact of these possible treatment modalities on the homeostatic intestine. This is especially relevant given the pronounced expression of Grem1 by crypt-associated Pdgfra-low mesenchymal cells in the homeostatic intestine.

- In general, the selection of markers (and associated stainings) for the mesenchymal cell populations used by the authors to judge whether mesenchymal populations are changing in various conditions is rather minimal. Here, performing more in-depth studies e.g., by performing single-cell RNA sequencing of various conditions (e.g. Vil1-Grem1 mice, Vil1-Grem1 mice + Grem1 antibody treatment etc.) would allow for more refined conclusions regarding changes in cell populations.

- Given the problem with partial recombination, why are the experiments using Vil1-CreERT2;Fzd5fl/fl;Vil1-Grem1 and Vil1CreERT2;Lgr4fl/fl;Vil1-Grem1 even mentioned to begin with? The authors themselves point towards a lack of interpretability due to entire escaper ribbons. Furthermore, why was this combination of receptors chosen? It is true, that both are expressed in ectopic crypts. However, even if the partial recombination in the mouse models wouldn't be an issue compensation by the paralogs of Lgr4 and Fzd5 could potentially cause major issues in the interpretation of the obtained results and should at least be carefully analyzed. In addition, the authors chose to circumvent the recombination issue by studying R26-CreERT2;Lgr4fl/fl; Vil1-Grem1 mice. Once more the matched control of the same timepoints in a R26-CreERT2;Lgr4fl/fl mouse are missing and should be performed by the authors.

Minor comments:

- Can the authors elaborate on how the experiments with γ -secretase inhibitor are relevant and connected to the ectopic Grem1 expression? They do not observe any effect of the inhibitor treatment on ectopic crypts and their morphology. As Olfm4 is Notch target is it not surprising that its expression is reduced upon the inhibitor treatment. The same can be said concerning goblet cells (alcian blue staining), as affecting the Notch pathway imbalances the cell ratio between enterocytes and secretory cells.

- Furthermore, the title of the manuscript is misleading, as it is not the endogenous mesenchyme derived Gremlin1 expression per se that disrupts epithelial-mesenchymal crosstalk, but the ectopic epithelial overexpression of Gremlin1. The title should be adjusted accordingly: "Ectopic epithelial overexpression of Gremlin1 disrupts intestinal epithelial-mesenchymal crosstalk...".

- Figure3C: This panel is highly confusing due to the overlap of the staining with computational regions (heatmap like areas). The real staining for individual probes (channels) should be shown, including quantifications (normal crypt, villus, aberrant crypt). Given its current obscurity, the display does not allow to agree or disagree with authors. The panels shown in Fig.S3A depict (most likely – not specified) wild type intestine, but raise the questions concerning the sensitivity of probes (for example for Pdgfra or Cd55).

Reviewer #3

(Remarks to the Author)

This work uses multiple in vivo models to elegantly demonstrate the effect of ectopic intestinal epithelial gremlin expression on disrupting the BMP gradient leading to alteration in the stromal niche that support stem cell activity. Suppressing gremlin expression of wnt signaling reverts this phenotype, which is also associated with altered Notch signaling. However, Notch inhibition promotes goblet cell differentiation and abrogates *Olfm4* expression but fails to revert ectopic crypt formation in the context of aberrant gremlin. More advanced tumors lost the dependency on wnt signaling and are refractory to porcupine inhibitors associated with accumulation of additional pro-tumorigenic alterations. Stromal alterations similar to that observed in the Gremlin overexpression mouse model (with aberrant accumulation of CD55(+) WNT2b(+) fibroblasts) are also found in HMPS polyps and sporadic TSA lesions, a proportion of which have marked ectopic crypts with accumulation of Gremlin. Together these results could be clinically relevant, at least for HMPS patients or other subsets of patients carrying tumor with gremlin accumulation.

Version 1:

Reviewer comments:

Reviewer #1

(Remarks to the Author)

The authors have adequately addressed all the points I raised in my previous review. I believe the paper is now ready for publication.

Reviewer #2

(Remarks to the Author)

I would like to thank the authors for their effort to address my comments. Despite the newly added data and increased clarity I am afraid there are still some unresolved questions and concerns that should be addressed.

1. The key questions, raised in the initial review, of how ectopic crypts are formed and how a supportive ectopic (remodeled) stem-cell niche appears remains unresolved. Does it occur through the remodeling of differentiated cells within the villus and the transition of the differentiated niche toward a stem cell niche, or is the first event the repression of Bmp-mediated differentiation in villi, followed by the expansion of crypt compartments and the formation of ectopic crypts? Apparently, *Grem1* begins to be aberrantly expressed postnatally in the case of vil-*Grem1*, but there are no ectopic crypts at p10. How are the epithelium and mesenchymal niche remodeled between p10 and p120? Or what happens in the case of inducible *Grem1* ectopic expression at earlier time points (from the induction until the appearance of the first ectopic crypts)? Will the first aberrant progenitor cells (e.g., Sox9+, Ki67+) appear in the villus? Is the niche remodeled first? Sox9 tracing was done when the ectopic crypts were already established; hence, it does not explain how they are formed. What is the mechanism behind this?

2. The figure showing the quantification of goblet and Ee cells in the point-by-point reply (R4) is not in agreement with the scRNAseq data. Whereas R4 does not point to any difference, the scRNA-seq data in Fig.S3B indicates an increase in both goblet cells and Ee cells in vil-*Grem1* epithelium. Surprisingly there is an apparent increase in number of enterocytes. This is counterintuitive, as the appearance of ectopic crypts, which contain mainly progenitor cells, seems to be at the expense of villus enterocytes. Please explain these discrepancies.

3. It will be very helpful to show mixed Lyz1+/Muc2+ cells are present in aberrant crypts. Ideally by co-immunostaining or, minimally, to visualize these double-positive cells more clearly in the tSNE plots (i.e. specifically indicate double positive cells in Muc2 and Lyz1 plots).

4. The visualisation of scRNAseq data should be extended and clarified. In the tSNE plot in Fig.3 A control, vil-*Grem1*, and UCB are all integrated into one plot. It would be beneficial to show not only the combined plot but also the individual parallels. This would allow the reader to visualize cellular populations and their changes under different conditions. Ideally, for each experimental parallel crypt and villus fractions could be distinguished. Violin plots or bar graphs are not so informative. Is there any specific cell type/population for vil-*Grem1* epithelium? In other words, how similar or different are ectopic crypts compared to healthy ones? Do the gene signatures of differently expressed genes in epithelial cell populations suggest differences in signals provided by surrounding non-epithelial cells?

5. Fig. 3A indicates an increase not only in Sox9+ cells but also in *Lgr5*+ cells. Does this mean there are any *Lgr5*+ cells within ectopic crypts that are not further mutated (e.g., mutation in β -catenin)?

6. Are ectopic crypts devoid of any *Lgr5* or other classical stem cell markers (*Olfm4*, etc.)? The authors claim that the *Lgr5*+ cells harbor further mutations (Fig. 6D, E), but is this always the case? What if ectopic crypts contain classical stem cells that are lost during polyp formation? In the point-by-point reply, the authors nicely showed (Fig. R6, mentioned only in the table) the potential of villi derived from vil-*Grem1* animals to form organoids. Do they express *Lgr5*? I strongly recommend adding a

simplified version of figure (R6) to the manuscript. The observation that vil-Grem1 villi give rise to organoids only in ENSW medium is interesting and important. To simplify the figure, one could omit combinations with Apc-min. It is known that loss of Apc can form ectopic crypts (doi: 10.1126/sciadv.abj0512, doi: 10.1038/s41598-018-38310-y).

7. It would be worthwhile to add representative panels showing individual stainings of Hi-PLEX ISH for vil-Grem1 (similarly to Fig. S4A) – or at least include the staining as shown in the point-by-point reply Fig. R7 (please indicate the color code for each panel). If possible, please clarify the color code for Fig. 4C in a way that clearly indicates which probes/Abs are used for each panels.

Version 2:

Reviewer comments:

Reviewer #2

(Remarks to the Author)

The authors addressed all my concerns sufficiently. I believe the manuscript is now suitable for acceptance. However, I would like to ask the authors to add the information about the limitations of their model either to the text or to the figure legend(s), as they stated in their rebuttal (.....The model is thus speculative, and we cannot know the exact cause of ectopic crypt initiation). I would like to congratulate the authors for an interesting manuscript.

Response to referees for: GREMLIN1 disrupts intestinal epithelial-mesenchymal crosstalk to induce a wnt-dependent ectopic stem cell niche via stromal remodelling" (reference number: NCOMMS-24-21207-T

Dear Editor,

Thank you for the thorough and constructive feedback provided by the reviewers on our manuscript. We greatly appreciate the reviewers' insights and in response we have undertaken considerable additional work, which we believe has strengthened the manuscript. We enclose a point-by-point response to the reviewers' comments and have highlighted all changes to the manuscript in red.

Reviewer 1

Leedham lab has reported a role of Gremlin1 in stromal remodeling. They developed a therapeutic strategy using a Gremlin1 antibody and a Porcupine inhibitor that prevented ectopic crypt formation in the hereditary mixed polyposis syndrome (HMPS) mouse model. In addition, the authors found that mesenchymal gene expression patterns were altered by epithelial Gremlin1 production, which in turn supported de novo stem cell niche environments and ectopic crypt formation. These results are interesting and appropriate for publication in Nature Communications.

We thank the reviewer for their constructive review and recognition of the value of the work

There are some issues that need to be addressed before its publication.

1. While they provide substantial data, some are not directly relevant to the main story. Specifically, the section on Notch signaling (ATOH1-CreER tracing and DBZ (Notch inhibitor) treatment) seems unnecessary (it is not included in the model summary of Figure 6c) and complicates the story. I suggest they either remove these parts or provide a more coherent explanation of why the Notch pathway is important in stromal remodeling.

The superfluous aspect of the Notch section of the paper was noted by 2 reviewers. We agree with the reviewer's comment and have removed these data which included Figure 5E-5G and Supplementary Figure 5F. We believe this has helped streamline the story.

2. The authors provided almost no mechanism of how epithelial Gremlin1 remodels the stromal niche environments. The epithelial-mesenchymal interactions illustrated in Figure 6c are interesting but somewhat hypothetical, as no biological data supports this. It would be beneficial if they could provide some functional data, such as in vitro remodeling of stroma by Gremlin1.

The Shivdasani lab has already undertaken the suggested experiments and published their results in Kraiczky et al. ¹ They isolated CD55+ fibroblasts from mouse intestines by FACS and cultured them in the presence of BMP ligands or inhibitors. These results, cited in our manuscript, show that *in vitro* BMP

manipulation can alter fibroblast phenotype, with media recombinant BMP antagonism boosting CD55 expression in these cells.

Figure R1. Data from Kraiczy *et al.* showing that BMP manipulation with recombinant BMP ligand (Green) or recombinant BMP antagonist (purple) alters fibroblast cell expression profiles *in vitro*.

In light of the reviewers comments we looked to replicate this Shivdasani data by extracting and culturing primary mouse fibroblasts for 7 days to assess response to *Grem1*. However, without FACS separation of stromal cell subtypes, we found that culturing primary mouse fibroblasts resulted in a loss of stromal cell functional heterogeneity, with *Grem1* and *Rspo3* expressing cells outcompeting other fibroblasts, to result in a homogenous trophocyte cell population in culture (Figure R2).

Figure R2. Real-time PCR results for primary murine fibroblasts, comparing post-harvest and 7 day cultured levels of *Grem1* and *Rspo3* expression indicating trophocyte outgrowth *in vitro*.

Consequently, adding additional media *Grem1* to these cells had no impact on fibroblast expression. However, we did undertake the reverse experiment, adding media BMP ligand, to confirm that fibroblasts are truly BMP responsive (Figure R3).

Figure R3. Gene expression results for primary murine fibroblasts cultured with recombinant *Bmp4* protein compared to standard conditions. *Bmp4* increased expression of the BMP target gene *Id1* and reduced trophocyte gene score.

These data highlight important limitations of working with fibroblasts in culture. Stromal cell behaviour and molecular phenotype is heavily influenced by their cellular environment and mechanical cues, which makes it tough to maintain functional diversity once we start culturing them on plastic. As such we believe that our approach to identify and spatially map these cell populations *in vivo*, within their native

environment is the best approach. To do this we designed a bespoke HiPlex RNA ISH panel marker panel based on the Kraiczky et al fibroblast cell markers, and we believe this convincingly demonstrates a reproducible response to aberrant epithelial *Grem1* expression which we hope the reviewer will find compelling.

They discussed FGFR activation by Gremlin1, but I would like to see if this is validated in their model.

Cheng *et al* demonstrated that Gremlin1 binds to FGFR1 and activates downstream MAPK signaling in prostate cancer ². In order to answer the reviewers query, we undertook assessment of MAPK activation in *Vil1-Grem1* lesions using IHC to visualise pMAPK (**Figure R4**). This revealed patchy expression of pMAPK in ectopic crypts within polyps, suggesting that there is more than direct *Grem1* expression which influences this change in intestinal tissue.

Figure R4. pMAPK expression shown using IHC in *Vil1-Grem1* polyp and adjacent normal. Aberrant epithelial expression appears to lead to patch activation of pMAPK in this model, with no direct correlation between epithelial *Grem1* expression and pMAPK observed.

Perhaps more interesting and relevant for the intestinal stem cell field was the very recent demonstration that *Fibroblast growth factor receptor binding protein 1 (Fgfbp1)* marks a putative population of upper

crypt stem cells^{3,4} and we have added some new data to address this. We stained both wild-type and *Vil1-Grem1* tissue for *Fgfbp1*, and found significant expression in the ectopic crypts, overlapping with *Sox9* expression, consistent with the aberrant expansion of an *Lgr5*-ve stem/progenitor cell population. This change was reversible with anti-Grem1 antibody treatment. We've now included these findings in Figures 1, 2, and S1 of the revised manuscript. Together the convergence of Grem1 and Fibroblast Growth Factor signalling is intriguing and is worthy of significant further study, however we believe this warrants a considerable amount of work that is beyond the scope of this current manuscript.

3. The Gremlin1 antibody treatment experiment elegantly demonstrated its therapeutic potential for HMPS mouse model. However, the authors treated the mice from 1-week old onwards, which only proved its preventative effect on tumorigenesis. I wonder if they could demonstrate whether polyps could be eliminated when treatment is started from 1-month old mice. Therapeutic data would significantly enhance the value of the antibody treatment strategy.

The manuscript did enclose data examining the timed effect of up to 10 weeks antibody therapy on 120 day old mice with established polyposis (Fig 2B,D,E and S2), however we accept that the distinction of this from long-term preventative antibody administration was not at all clear in the original manuscript. We have improved this in the text and the revised figure 2, clearly separating out the preventative and treatment arms of antibody use.

Reviewer 2

*In this manuscript Mulholland and Belnoue-Davis and colleagues describe the impact of constitutive pan-epithelial Gremlin 1 over-expression on intestinal homeostasis. In particular, they describe the formation of ectopic epithelial crypts, reminiscent of HMPS-phenotypes, and the role of stromal cells in the establishment of this Gremlin1-induced phenotype. It is an extension of previous work from the same lab (<https://doi.org/10.1038/nm.3750>) that already describes the formation of ectopic crypts in this mouse model (*Vil1-Grem1* mice). Here, the authors add data regarding the potential underlying mechanisms, rooted in the pharmacological reversion of the phenotype using a Gremlin1-antibody (UCB Ab7326). This finding is interesting and adds potential translational value for HMPS patients. However, the manuscript remains rather descriptive and superficial with important controls missing, as outlined below in the point-by-point.*

We thank the reviewer for their informed and comprehensive review.

• The authors do not analyze the fate of cells that depend on Bmp signaling – in particular enterocytes and enteroendocrine cells (as shown in publications by Beumer and Clevers: doi: 10.1016/j.celrep.2022.110438, doi: 10.1016/j.cell.2020.08.005) in both their Vil1-Grem1 mouse model, as well as upon Grem1 antibody (UCB Ab7326) treatment. This is a major omission and requires critical further analysis. Along similar lines, in the text corresponding to Figure 1I the authors claim: “... lead to disrupted secretory cell fate”. What exactly do the authors mean by this? They did not analyze secretory markers and they did not perform extensive expression profiling (e.g., single-cell RNA sequencing). As

mentioned, Beumer and Clevers showed that Bmp signaling may affect the cell fate of goblet cells (e.g., the production of antimicrobial peptides). Therefore, as Gremlin1 inhibits Bmp signaling is goblet cell fate disrupted in Vil1-Grem1 mice?

We appreciated this comment, as it helped us recognise that the take home point of this experiment was not being adequately communicated. The aim of this section was to contrast the effect of aberrant epithelial Grem1 expression on cell fate when expressed either at single cell or pan-epithelial level. This allows us to demonstrate that pan-epithelial expression results in ectopic stem cell behaviour but that this is not induced intrinsically when *Grem1* is expressed in individual cells. We used *Atoh-CreER^{T2}* to induce expression in individual secretory progenitor cells, as it is a model we have experience with and know well. Consequently, this was never intended to be a study of the impact of BMP signalling on secretory cell fate which has already been comprehensively covered in the Clevers lab publications the reviewer highlights. We have re-written this section, changed the pan-epithelial expression model (see later comment) and clarified the key findings from the experiment.

In light of the reviewers comments, including those on secretory cell fate, we have also undertaken single cell analysis of UCB Ab7326 treated and untreated *Vil1-Grem1* mouse model, and enclose the data from this in a new Figure 3 and S3. As requested, we have analysed secretory cell phenotypes transcriptionally and spatially (see data below) and only find quantitative differences in Lysozyme/Muc2 co-expressing cells (Fig S3), which aberrantly appear in ectopic crypts and on the villi (Fig 2) , as previously noted ⁵. Given the previous findings from the Clevers lab, these cells may represent an impact of pan-mucosal BMP gradient disruption on Paneth/goblet cell maturation in *Vil1-Grem1* animals.

A

Goblet cells (Alcian Blue)

Wild-type

Vil1-Grem1

B

Enteroendocrine cells (CgA)

Wild-type

Vil1-Grem1

Figure R4. Quantitative analysis of goblet and enteroendocrine secretory cell proportions in normal and *Vil-Grem1* mouse models. Goblet cells identified using Alcian Blue staining, and Enteroendocrine cells identified using IHC for CgA. Quantification was conducted using QuPath Image Analysis Software.

• Another major aspect that is fully ignored by the authors is the timing of *Grem1* expression in their *Vil1-Grem1* mouse model. Given the nature of the mouse model, the expression of *Grem1* is not inducible. As *Vil1* starts to be expressed during embryonic development it is highly likely that already the development of the embryonic gut is altered and proper intestinal niches are not established to begin with, which severely affects the interpretation of the presented phenotypes. What if e.g., *Grem1*-induced ectopic crypts retained embryonic features due to a corrupted development (in *Vil1-Grem1* animals)? The authors should investigate such issues using proper time-course analysis starting already during embryonic gut development.

This is a very interesting question that we have carefully considered and we now enclose data that we hope will reassure the reviewer. To address these questions we harvested embryonic and post-natal gut tissue from wild-type and *Vil1-Grem1* mice at E13.5 (data not shown), E16.5, P0.5, and P10. Villin starts to be expressed from E9.5 and so these time points allow us to explore cell dynamics influenced by epithelial *Grem1* in the embryonic gut. Staining for stem cell markers we found no significant change or difference in stem cells between the wild-type and *Vil1-Grem1* genotypes (new results section and Figure S1) Thus it appears that the impact of aberrant ectopic epithelial *Grem1* only manifests itself in adult mice resulting in an observable phenotype in animals from about 3 months of age.

The *Vil1-Grem1* model was designed to phenocopy Hereditary Mixed Polyposis Syndrome which is a germline, autosomally dominant inherited condition. Interestingly the HMPS disease phenotype is also only seen in adult patients, with a mean age of clinical presentation of 40 (as opposed to Familial Adenomatous Polyposis which can present in childhood). How, and why the developing gut is able to apparently buffer aberrant expression of a BMP antagonist is an interesting question that merits further expert study. However, we hope these new data will address the reviewer's immediate concerns over adult mouse phenotype interpretation.

• Their choice of model system leads to additional interpretative issues when the authors attempt to distinguish between the effects of Grem1 overexpression from individual secretory cells vs. the entire epithelium. Their comparison between Atoh1-CreERT2;R26-LSL-Grem1 and Atoh1-CreERT2;R26-tdTom;Vil1-Grem1 is not only a comparison between expression of Grem1 from individual secretory cells vs. pan-epithelial expression, but more importantly a comparison between acute overexpression of Grem1 in the adult intestinal context vs. constitutive Grem1 overexpression for the entire developmental history (including embryonic development) of the intestine. As such, their interpretation of the observed differences in phenotype are misleading. It would be more appropriate to compare e.g., Vil1CreERT2;R26-LSL-Grem1 (pan-epithelial) with Atoh1CreERT2;R26-LSL-Grem1 (secretory specific) mice to support the authors claims.

We predominantly use the *Vil1-Grem1* mouse in the paper, as it is a disease-positioned model of the human germline HMPS condition, however we do agree with the reviewer's comment that it was not the appropriate model for the individual cell vs pan-epithelial expression experimental question. In line with the reviewer's suggestion we have substituted *Vil1CreERT2;Rosa26^{Grem1}* in this section to explore the phenotypic impact of inducible pan-epithelial expression of *Grem1* on adult tissue (Figure 1).

• Why did the authors choose Sox9 to trace the origin of the ectopic crypts and why was only partial recombination induced to trace this lineage? This can lead to incorrect observations/statements as some relevant Sox9 progenitors might not be labeled. The essential control condition however is missing – tracing Sox9 for the same timeframes in the wildtype/homeostatic intestine.

Sox9 was chosen for tracing as it is a known stem/progenitor cell marker in the intestine⁶⁻⁸ and is strongly expressed in *Lgr5*- ectopic crypt cells in *Vil1-Grem1*⁵. We now show additional data demonstrating co-

expression of *Sox9* and *Fgfbp1*, with the latter marking a putative population of upper crypt stem cells^{3,4} that appear to be expanding within *Vil1-Grem1* mice ectopic crypts (Fig 1A and Fig S1A).

In light of the known co-expression of *Sox9* in the TA cells of the normal crypt we wished to avoid widespread crypt basal recombination which could have obscured desired lineage tracing from the villus ectopic crypt cells. We thus followed the low dose recombination protocol established and published by Doug Winton's group^{9,10}.

The lineage tracing capability of *Sox9*+ cells from homeostatic crypt basal cells with the same animal model has been previously published^{8,11}, however we enclose the requested images of crypt basal tracing in a wild-type setting (Figure R5).

Figure R5. *Sox9-CreER^{T2}; Rosa26^{YFP}* with anti-YFP staining using IHC at 24 h and 7 days post-recombination, showing crypt base cell lineage tracing ribbons.

The authors suggest that Sox9+(Lgr5-) cells in the Vil1-Grem1 mice have stem cell potential in the ectopic crypts. It would be nice if they could show this e.g., by isolating these cells and test their ability to form epithelial organoids (as has previously been shown for Lgr5+ cells isolated from the homeostatic intestine).

Here, we have demonstrated the stem cell potential of ectopic crypt *Sox9*+ cells using a gold standard lineage tracing experiment *in vivo*, and have previously shown the capacity of *Vil1-Grem1* mouse villi to generate organoids (Figure R6)⁵, so we feel that this point has been comprehensively covered.

Media supplementation	Mouse genotype and tissue compartment						
	Wild-type crypts	Vil1-Grem1 crypts	Apc ^{Min} polyps	Wild-type villi	Apc ^{Min/+} villi	Vil1-Grem1 villi	Vil1-Grem1/Apc ^{Min} villi
ES		Branching enteroid	Spheroid				Spheroid
ES + rBMP2,4,7			Spheroid				Spheroid
ENS	Branching enteroid	Branching enteroid	Spheroid				Spheroid
ENSW	Branching enteroid	Branching enteroid	Spheroid			Spheroid	Spheroid

Figure R6. Organoid forming potential of villi in *Vil1-Grem1* and derivative models (from Davis *et al*). This data shows that *Vil1-Grem1* villi can form organoids under standard media conditions.

• As stated by the authors, a correct balance of *Wnt*- and *Bmp* gradients are integral to the establishment of the intestinal stem cell and differentiation niches. As such, it is very surprising that treatment with *Grem1* antibody (UCB Ab7326) or Porcupine inhibition (LGK-974) result only in a reversal of the aberrant phenotypes in *Vil1-Grem1* animals and do not affect homeostatic niches and as a result mouse survival at the same time. Here, again important control experiments are missing. The authors should test these treatment modalities in wildtype mice to judge the impact of these possible treatment modalities on the homeostatic intestine. This is especially relevant given the pronounced expression of *Grem1* by crypt-associated *Pdgfra*-low mesenchymal cells in the homeostatic intestine.

We apologise that these data were not in the manuscript and have included the relevant treated wild-type groups for both UCB Ab7326 and LGK-974 in Figures S2 and S5 respectively. Interestingly, both drugs have convergent effects in reducing the intensity of expression of *Lgr5* in the crypt base stem cells, albeit through different mechanisms. With regards to LGK-974, this is a dose-dependent effect with higher doses resulting in stem cell loss and intestinal failure (as previously published¹²), however with UCB Ab7326 even high doses of antibody had no overall effect on mouse survival, indicating potential BMP antagonist redundancy.

• In general, the selection of markers (and associated stainings) for the mesenchymal cell populations used by the authors to judge whether mesenchymal populations are changing in various conditions is rather minimal.

The custom 12-marker Hi-Plex panel was specifically designed to identify the different fibroblast cell phenotypes identified and characterised in intestinal tissue by the Shivdasani group ^{1,13,14}. It is a more comprehensive spatial panel than that published previously, as we triangulate co-expression of 2-3 spatial transcriptome markers for each cell phenotype ^{1,13-15}. The panel was also sufficient for the objective of comprehensively demonstrating the de-repression and expansion of the Wnt2b, Cd55+ stromal cell population in the *Vil1-Grem1* model. The Hi-Plex technology used was state-of-the-art at the time of the experiment (early 2023), and utilised the maximum simultaneous marker multiplexing that could be achieved then. As such, we do not believe that mesenchymal marker selection used was insufficient in allowing for the characterization of known stromal cell populations, or that a broader marker panel (now subsequently possible through new single cell transcriptomic technologies like Xenium or CosMx) would be any more effective in addressing the scientific question.

*Here, performing more in-depth studies e.g., by performing single-cell RNA sequencing of various conditions (e.g. *Vil1-Grem1* mice, *Vil1-Grem1* mice + *Grem1* antibody treatment etc.) would allow for more refined conclusions regarding changes in cell populations.*

We have undertaken scRNA expression of the suggested mouse groups and the data is included in the manuscript and new Figure 3. Unfortunately, isolation of stromal cell populations for single cell experiments is notoriously difficult leading to frequent under-representation of these cell populations in single cell datasets ¹⁶. Despite following optimised extraction protocols, the yield of stromal cells here was not sufficient to explore single cell transcriptome fibroblast heterogeneity with any granularity. This, together with the ability to visualise spatially resolved cells within the context of the tissue is why we favour the use of advancing spatial biology technologies in stromal cell phenotyping ¹⁷.

*• Given the problem with partial recombination, why are the experiments using *Vil1-CreERT2;Fzd5fl/fl;Vil1-Grem1* and *Vil1CreERT2;Lgf4fl/fl;Vil1-Grem1* even mentioned to begin with? The authors themselves point towards a lack of interpretability due to entire escaper ribbons. Furthermore, why was this combination of receptors chosen? It is true that both are expressed in ectopic crypts. However, even if the partial recombination in the mouse models wouldn't be an issue, compensation by the paralogs of *Lgr4* and *Fzd5* could potentially cause major issues in the interpretation of the obtained results and should at least be carefully analyzed.*

We originally included the long term *Lgr4* and *Fzd5* knockout models as we felt they demonstrated a strong selective pressure for escaper crypts highlighting the importance of these receptors in ectopic crypt wnt signal transduction. However, we accept that these experiments are correlative and are superseded by the findings of the acute knockout of *Lgr4* in the *Rosa26-CreER^{T2};Lgr4^{fl/fl}; Vil1-Grem1* model.

Consequently we agree with the reviewers point and have removed the long-term knockout models to improve manuscript clarity.

In addition, the authors chose to circumvent the recombination issue by studying R26-CreERT2;Lgr4fl/fl; Vil1-Grem1 mice. Once more the matched control of the same time points in a R26-CreERT2;Lgr4fl/fl mouse are missing and should be performed by the authors.

We respectfully disagree here. The experiment is to show the effect of acute knockout of Lgr4 on ectopic crypt proliferation in *Vil1-Grem1* animals. Consequently, we show both recombined and control (non-recombined) *Vil1-Grem1* animals in Figure 6.

The effect of Lgr4 knockout in *R26-CreERT2;Lgr4^{fl/fl}* animals is a different experimental question concerning the effect of Lgr4 in steady-state (i.e. non *Vil1-Grem1*) conditions. We have these data as part of other experiments but do not feel that they are necessary to interpret the specific experiment included here.

Minor comments

*• Can the authors elaborate on how the experiments with γ -secretase inhibitor are relevant and connected to the ectopic Grem1 expression? They do not observe any effect of the inhibitor treatment on ectopic crypts and their morphology. As *Olfm4* is Notch target is it not surprising that its expression is reduced upon the inhibitor treatment. The same can be said concerning goblet cells (alcian blue staining), as affecting the Notch pathway imbalances the cell ratio between enterocytes and secretory cells.*

In agreement with this comment from two reviewers we have removed the Notch signalling pathway portion of the story which included Figure 5E-5G and Supplementary Figure 5F.

• Furthermore, the title of the manuscript is misleading, as it is not the endogenous mesenchyme derived Gremlin1 expression per se that disrupts epithelial-mesenchymal crosstalk, but the ectopic epithelial overexpression of Gremlin1. The title should be adjusted accordingly: "Ectopic epithelial overexpression of Gremlin1 disrupts intestinal epithelial-mesenchymal crosstalk..."

We have adjusted the manuscript title accordingly.

*• Figure3C: This panel is highly confusing due to the overlap of the staining with computational regions (heatmap like areas). The real staining for individual probes (channels) should be shown, including quantifications (normal crypt, villus, aberrant crypt). Given its current obscurity, the display does not allow to agree or disagree with authors. The panels shown in Fig.S3A depict (most likely – not specified) wild type intestine, but raise the questions concerning the sensitivity of probes (for example for *Pdgfra* or *Cd55*).*

We respectfully rebut this comment and enclose our reasons why. Visual quantitative analysis of tissue sections has previously been confounded by inevitable researcher subjectivity, with selection of images and cell calling providing a degree of conscious or unconscious bias. Digital pathological analysis using the

HALO digital pathology software allows us to effectively segment cells, assign marker positivity in an automated fashion and use digital masking to exclude unwanted stained epithelial cells. This technology generates unbiased quantitative analysis, providing an increased reassurance in the validity of the results. It is also capable of demonstrating the abundance and distribution of cells in an easily visualised, reader-friendly heatmap format.

With the tissue resolution of the images in Fig3C, non-zoomable unmarked tissue sections (Figure R7) provide very little additional information, and are not able to demonstrate the key point - the considerable villus expansion of the de-repressed Wnt2b+,CD55+ cells.

However, we agree that the generated quantification from the software is useful and have moved this from the supplementary to the main figure as requested.

Figure R7. Hi-plex RNAScope staining of non-zoomable unmarked sections for both Wild-type (Top) and *Vil1-Grem1* (Bottom) tissue samples. This demonstrates the value of the density plots in the main figure in allowing the reader to see the distribution and relative abundance of the different fibroblast subsets.

Reviewer 3

This work uses multiple in vivo models to elegantly demonstrate the effect of ectopic intestinal epithelial gremlin expression on disrupting the BMP gradient leading to alteration in the stromal niche that supports

stem cell activity. Suppressing gremlin expression of wnt signaling reverts this phenotype, which is also associated with altered Notch signaling. However, Notch inhibition promotes goblet cell differentiation and abrogates Olfm4 expression but fails to revert ectopic crypt formation in the context of aberrant gremlin. More advanced tumors lost the dependency on wnt signaling and are refractory to porcupine inhibitors associated with accumulation of additional pro-tumorigenic alterations. Stromal alterations similar to that observed in the Gremlin overexpression mouse model (with aberrant accumulation of CD55(+) WNT2b(+) fibroblasts) are also found in HMPS polyps and sporadic TSA lesions, a proportion of which have marked ectopic crypts with accumulation of Gremlin. Together these results could be clinically relevant, at least for HMPS patients or other subsets of patients carrying tumor with gremlin accumulation.

We thank the reviewer for their kind comments and especially appreciate their recognition of the value of this work for HMPS patients.

We believe these revisions have significantly strengthened the manuscript and comprehensively addressed the reviewers' concerns with a large volume of additional experimental work. We are pleased to submit the improved version for your reconsideration. Please let us know if you have any other questions or require additional information.

Sincerely,

Eoghan Mulholland
Hayley Belnoue-Davis
Simon Leedham

References

- 1 Kraiczy, J. *et al.* Graded BMP signaling within intestinal crypt architecture directs self-organization of the Wnt-secreting stem cell niche. *Cell Stem Cell* **30**, 433-449.e438 (2023). <https://doi.org:10.1016/j.stem.2023.03.004>
- 2 Cheng, C. *et al.* Gremlin1 is a therapeutically targetable FGFR1 ligand that regulates lineage plasticity and castration resistance in prostate cancer. *Nat Cancer* **3**, 565-580 (2022). <https://doi.org:10.1038/s43018-022-00380-3>
- 3 Capdevila, C. *et al.* Time-resolved fate mapping identifies the intestinal upper crypt zone as an origin of Lgr5+ crypt base columnar cells. *Cell* **187**, 3039-3055.e3014 (2024). <https://doi.org:10.1016/j.cell.2024.05.001>
- 4 Malagola, E. *et al.* Isthmus progenitor cells contribute to homeostatic cellular turnover and support regeneration following intestinal injury. *Cell* **187**, 3056-3071.e3017 (2024). <https://doi.org:10.1016/j.cell.2024.05.004>
- 5 Davis, H. *et al.* Aberrant epithelial GREM1 expression initiates colonic tumorigenesis from cells outside the stem cell niche. *Nat Med* **21**, 62-70 (2015). <https://doi.org:nm.3750> [pii] 10.1038/nm.3750

- 6 Formeister, E. J. *et al.* Distinct SOX9 levels differentially mark stem/progenitor populations and enteroendocrine cells of the small intestine epithelium. *Am J Physiol Gastrointest Liver Physiol* **296**, G1108-1118 (2009). <https://doi.org:00004.2009> [pii]
10.1152/ajpgi.00004.2009
- 7 Ramalingam, S., Daughtridge, G. W., Johnston, M. J., Gracz, A. D. & Magness, S. T. Distinct levels of Sox9 expression mark colon epithelial stem cells that form colonoids in culture. *Am J Physiol Gastrointest Liver Physiol* **302**, G10-20 (2012). <https://doi.org:ajpgi.00277.2011> [pii]
10.1152/ajpgi.00277.2011
- 8 Furuyama, K. *et al.* Continuous cell supply from a Sox9-expressing progenitor zone in adult liver, exocrine pancreas and intestine. *Nat Genet* **43**, 34-41 (2011). <https://doi.org:ng.722> [pii]
10.1038/ng.722
- 9 Lopez-Garcia, C., Klein, A. M., Simons, B. D. & Winton, D. J. Intestinal stem cell replacement follows a pattern of neutral drift. *Science* **330**, 822-825 (2010). <https://doi.org:science.1196236> [pii]
10.1126/science.1196236
- 10 Tomic, G. *et al.* Phospho-regulation of ATOH1 Is Required for Plasticity of Secretory Progenitors and Tissue Regeneration. *Cell Stem Cell* **23**, 436-443.e437 (2018). <https://doi.org:10.1016/j.stem.2018.07.002>
- 11 Roche, K. C. *et al.* SOX9 Maintains Reserve Stem Cells and Preserves Radioresistance in Mouse Small Intestine. *Gastroenterology* **149**, 1553-1563.e1510 (2015). <https://doi.org:https://doi.org/10.1053/j.gastro.2015.07.004>
- 12 Huels, D. J. *et al.* Wnt ligands influence tumour initiation by controlling the number of intestinal stem cells. *Nat Commun* **9**, 1132 (2018). <https://doi.org:10.1038/s41467-018-03426-2>
- 13 McCarthy, N., Kraiczy, J. & Shivdasani, R. A. Cellular and molecular architecture of the intestinal stem cell niche. *Nat Cell Biol* **22**, 1033-1041 (2020). <https://doi.org:10.1038/s41556-020-0567-z>
- 14 McCarthy, N. *et al.* Distinct Mesenchymal Cell Populations Generate the Essential Intestinal BMP Signaling Gradient. *Cell Stem Cell* **26**, 391-402.e395 (2020). <https://doi.org:10.1016/j.stem.2020.01.008>
- 15 McCarthy, N. *et al.* Smooth muscle contributes to the development and function of a layered intestinal stem cell niche. *Dev Cell* **58**, 550-564.e556 (2023). <https://doi.org:10.1016/j.devcel.2023.02.012>
- 16 Waise, S. *et al.* An optimised tissue disaggregation and data processing pipeline for characterising fibroblast phenotypes using single-cell RNA sequencing. *Sci Rep* **9**, 9580 (2019). <https://doi.org:10.1038/s41598-019-45842-4>
- 17 Bull, J. A. *et al.* Integrating diverse statistical methods to analyse stage-discriminatory cell interactions in colorectal neoplasia. *bioRxiv*, 2024.2006.2002.597010 (2024). <https://doi.org:10.1101/2024.06.02.597010>

Dear Editor,

Thank you for the thorough and constructive feedback provided by the reviewers on our manuscript entitled "**Epithelial GREMLIN1 disrupts intestinal epithelial-mesenchymal crosstalk to induce a wnt-dependent ectopic stem cell niche via stromal remodelling.**" We greatly appreciate the reviewers' insights and in response we have undertaken considerable additional work, which we believe has strengthened the manuscript. We enclose a point-by-point response to the reviewers' comments and have highlighted all changes to the manuscript in red.

Reviewer #1 (Remarks to the Author):

The authors have adequately addressed all the points I raised in my previous review. I believe the paper is now ready for publication.

We thank reviewer 1 (and reviewer 3) for their positive and constructive assessment of the manuscript and are glad that they feel it is publication ready.

Reviewer #2 (Remarks to the Author):

I would like to thank the authors for their effort to address my comments. Despite the newly added data and increased clarity I am afraid there are still some unresolved questions and concerns that should be addressed.

1. The key questions, raised in the initial review, of how ectopic crypts are formed and how a supportive ectopic (remodeled) stem-cell niche appears remains unresolved.

Our manuscript concerns the mechanism of formation and maintenance of ectopic crypts, with a number of novel findings that move the field away from an epithelial-centric paradigm by supporting a proposed epithelial-mesenchymal interdependence and co-evolution model.

Does it occur through the remodeling of differentiated cells within the villus and the transition of the differentiated niche toward a stem cell niche, or is the first event the repression of Bmp-mediated differentiation in villi, followed by the expansion of crypt compartments and the formation of ectopic crypts? Apparently, *Grem1* begins to be aberrantly expressed postnatally in the case of *Vil-Grem1*, but there are no ectopic crypts at p10. How are the epithelium and mesenchymal niche remodeled between p10 and p120? Or what happens in the case of inducible *Grem1* ectopic expression at earlier time points (from the induction until the appearance of the first ectopic crypts)? Will the first aberrant progenitor cells (e.g., *Sox9*+, *Ki67*+) appear in the villus? Is the niche remodeled first? *Sox9* tracing was done when the ectopic crypts were already established; hence, it does not explain how they are formed. What is the mechanism behind this?

Thank you for these further questions that concern the sequence of events that occur around the initiation of ectopic crypts. Whilst the *Vil1-Grem1* model has been extremely powerful in allowing us to understand this condition, it is not a model that is conducive to intravital imaging thus the dynamics

of ectopic crypt formation can only be inferred from temporally spaced snapshots. We now include the requested information from further timepoints (day p10 to p100) alongside data from the different morphological stages of polyp development and can use this information to speculate on the dynamics that occur at the point of ectopic crypt initiation.

In new data we document the impact that aberrant epithelial *Grem1* expression has on the expansion of epithelial cell number that occurs following the immediate postnatal period (**new Fig S1B**). This also profoundly increases overall intestinal size as seen in Davis *et al.* We now show that the proportion of Ck20 expressing differentiating cells reduces as lesions progress through key morphological stages (**new Figs S1C,D**). Together with what is known about the importance of an intact Bmp signalling gradient allows us to infer that aberrant epithelial *Grem1* expression increases the size of the postnatal epithelial cell compartment, probably through disruption of appropriate cell differentiation and shedding.

As the epithelial compartment expands in size, the stromal compartment is also undergoing concomitant remodelling. Looking at the different morphological stages of polyp development in *Vil1-Grem1* animals we see an initial expansion in the pericryptal population of Cd55, Wnt2b(+) cells which then extends onto the villi at time of emergence of ectopic crypts (**new Fig S5A,B**). These cells are then seen in large numbers, co-localising around ectopic crypts in established polyps, forming an important functional and reversible part of the ectopic niche, as detailed in the paper.

Given these new data and the important work of Shyer *et al*(1), which demonstrates that physical buckling of the epithelial layer in villus development alters the underlying stromal cell distribution to result in sub-epithelial stromal cell aggregations, we propose the following sequence of events:

1. Aberrant epithelial *Grem1* expression expands the postnatal epithelial cell compartment through altered cell differentiation and shedding (**Fig S1**).
2. The expanded epithelial cell compartment generates enlarged villi (**Fig S1**).
3. Disrupted Bmp signalling concomitantly de-represses the restricted pericrypt Cd55, Wnt2b(+) expressing cell population, which expands luminally, along the length of the crypt and up into the villi (**Fig S5**).
4. Physical buckling of the expanded villus epithelial cell layer generates invaginating intravillus ectopic crypts. (**Fig S1**)
5. Bi-directional, interdependent signalling between the invaginating ectopic crypt and the expanded Wnt expressing stromal cell population provides optimal conditions for aberrant, supra crypt stem cell functionality (in cells marked by Sox9, *Fgfbp1*, *Olfm4* expression).

We must stress that this is a model inferred from the increased number of temporal and morphological snapshots we now present and the available literature. The model is thus speculative, and we cannot know the exact cause of ectopic crypt initiation - for example whether epithelial invagination occurs through physical buckling of an expanded villus cell compartment (as proposed by Shyer *et al.*) or in response to an expanding underlying stromal wnt source. However, this concept is consistent with the idea that the formation of ectopic crypts is the consequence of Bmp signalling disruption on the interdependent and co-evolving epithelial and stromal cell compartments. We believe that this

important idea, which is the key and substantive message of the paper, should resonate widely in the intestinal stem cell biology field.

We have added these additional data (**Figure S1, S4**) and adjusted the text accordingly.

2. The figure showing the quantification of goblet and Ee cells in the point-by-point reply (R4) is not in agreement with the scRNAseq data. Whereas R4 does not point to any difference, the scRNA-seq data in Fig.S3B indicates an increase in both goblet cells and Ee cells in vil-Grem1 epithelium. Surprisingly there is an apparent increase in number of enterocytes. This is counterintuitive, as the appearance of ectopic crypts, which contain mainly progenitor cells, seems to be at the expense of villus enterocytes. Please explain these discrepancies.

We believe that the observed discrepancies between the single cell and the tissue staining predominantly relate to differences between the assessment of absolute cell numbers in the single cell dataset and the use of cell proportion which we use to quantify tissue staining. Absolute cell numbers are affected by the postnatal increase in the size of the crypt-villus unit and significant villus crowding that occurs in adult *Vil1-Grem1* animals (**New Fig S1B**) which generates a much higher tissue biomass per unit size of intestine (**Figure R1**). As such, the total number of epithelial cells analysed in the *Vil1-Grem1* mice are markedly higher than that in the wild-type.

Figure R1. Biomass difference between mice. H&E images of equivalent pieces of tissue in wildtype and *Vil1-Grem1* small intestine, illustrating profound differences in biomass due to villus expansion, villus crowding and polyp formation

With regards to the enterocyte population, we have generated new data that quantifies this increased biomass over time with the result that the absolute numbers of cells (including enterocytes) are increased in *Vil1-Grem1* mice because of the increase in crypt-villus unit size and villus crowding. This is despite a reduction in the proportion of Ck20+ cells that decreases with advancing morphological phenotype (as a consequence of ectopic crypt formation) (**New Fig S1C,D**).

We now present the total number of cells identified alongside the corresponding % of epithelial cells in the sample for each cell type in the revised **Figure 3B**. Here, it is evident that despite the increase in the total amount of enterocytes in the *Vil1-Grem1* mice pointed out by the reviewer, the percentage of the epithelial cells that are enterocytes between conditions do not differ as vastly. The expansion of the stem progenitor cell population in the *Vil1-Grem1* mice, and its reversion with UCB Ab7326 treatment continues to be evident when examined as proportion of epithelial cells.

The same metrics explain the reviewer's query pertaining to differences between goblet and EE cells. In Point 4 of the previous reply to reviewers, goblet cells and EE cells were evaluated from imaging, and were quantified as the percentage of cells. We have now adjusted the single cell data to show the percentage of epithelial cells in the sample for each cell type and we do not observe a substantial increase in goblet cells and EE cells as a proportion of epithelial cells (**Fig 3**), similar to the quantification from imaging previously shown.

3. It will be very helpful to show mixed Lyz1+/Muc2+ cells are present in aberrant crypts. Ideally by co-immunostaining or, minimally, to visualize these double-positive cells more clearly in the tSNE plots (i.e. specifically indicate double positive cells in Muc2 and Lyz1 plots).

We have added a visualization of Lyz1+(Muc2-) cells, Muc2+(Lyz1-) cells, and Lyz1+Muc2+ double positive cells in the scRNA-seq data. We observe expansions of the double positive population particularly in the goblet cells of *Vil1-Grem1* mice, which is substantially reversed in *Vil1-Grem1* mice treated with UCB Ab7326. These mixed goblet/paneth cell phenotypes can be visualised within the tissue context, by co-staining with Lysozyme and Alcian Blue (**Figure S3**)

4. The visualisation of scRNAseq data should be extended and clarified. In the tSNE plot in Fig.3 A control, vil-Grem1, and UCB are all integrated into one plot. It would be beneficial to show not only the combined plot but also the individual parallels. This would allow the reader to visualize cellular populations and their changes under different conditions. Ideally, for each experimental parallel crypt and villus fractions could be distinguished.

We have added to the visualization of the scRNA-seq data as t-SNE plots to include parallel t-SNE plots distinguishing the cells for each condition and crypt/villus fraction as requested by the reviewer.

Violin plots or bar graphs are not so informative. Is there any specific cell type/population for vil-Grem1 epithelium? In other words, how similar or different are ectopic crypts compared to healthy ones? Do the gene signatures of differently expressed genes in epithelial cell populations suggest differences in signals provided by surrounding non-epithelial cells?

We have revisited the scRNAseq data and re-clustered the Stem progenitor cells and the Enterocytes, which are the largest populations represented in the dataset, and for which there are enough cells measured to allow for common patterns of gene expression within the cell populations to be characterized. We do not identify subclusters in the Stem progenitor cells nor the Enterocytes that are represented only in a single genotype/treatment condition (**Figure R2**) - i.e. we do not identify any specific cell type/population for the *Vil1;Grem1* epithelium, as we observe expansions of subclusters in *Vil1;Grem1* epithelium that are represented/observed in wild-type tissue and after UCB Ab7326 treatment.

Figure R2. Subclustering of Stem progenitor cells and Enterocytes identifies clusters that exhibit expansions in certain mouse genotypes/conditions, but are not uniquely . Figure shows the number of cells in wild type, *Vil1;Grem1*, and *Vil1;Grem1* + UCB Ab7326 crypt and villi compartments for subclusters of (A) Stem progenitor cells and (B) Enterocytes.

5. Fig. 3A indicates an increase not only in Sox9+ cells but also in Lgr5+ cells. Does this mean there are any Lgr5+ cells within ectopic crypts that are not further mutated (e.g., mutation in β -catenin)?

In light of the acknowledged limitations of our single cell experiment - particularly our inability to completely exclude normal crypt contamination in the disaggregated villus fraction, we greatly favour interpretation of cell phenotypes using *ex vivo* spatial analysis where we can confidently segregate normal and ectopic crypts morphologically. When we examine the numbers of stem progenitor cells in the *Vil1;Grem1* and *Vil1;Grem1* + UCB Ab7326 with *Lgr5* expression > 0, while *Lgr5*+ cells can be observed in *Vil1;Grem1* villi in scRNA-seq data (in what is currently **Figure 3C**), these represent a very small quantity of cells that cannot be completely excluded as normal crypt contamination (**Figure R3**). Consequently, for this question about ectopic crypt stem cell phenotypes we refer the reviewer to the response to point 6.

Figure R3. Cells expressing *Lgr5* in stem progenitor cells from scRNA-seq data. Figure shows the number of cells in *Vil1;Grem1* and *Vil1;Grem1* + UCB Ab7326 crypt and villi compartments where *Lgr5* expression is > 0. *Lgr5*⁺ stem progenitor cells in the villus compartments in these mice represent a very small fraction of the *Lgr5*⁺ stem progenitor cells compared to what is observed in the crypts.

6. Are ectopic crypts devoid of any *Lgr5* or other classical stem cell markers (*Olfm4*, etc.)? The authors claim that the *Lgr5*⁺ cells harbor further mutations (Fig. 6D, E), but is this always the case? What if ectopic crypts contain classical stem cells that are lost during polyp formation?

This question has been comprehensively covered in our work. We refer the reviewer to our previous paper (Davis et al) where we assessed the expression of a range of classical stem cell markers, alongside the additional data already included in this manuscript (Fig 2). To summarise, ectopic crypt cells do not express *Lgr5* at visually detectable levels until the acquisition of a wnt disrupting mutation within morphologically advanced (high grade dysplasia) lesions. The expression of *Lgr5* (as a direct wnt target) is thus a consequence of mutation-acquired, cell-intrinsic wnt disruption.

Figure R4: *Lgr5* expression in different morphological stages of *Vil1-Grem1* polyp progression. *Lgr5* expression is restricted to the normal crypt bases in all stages of polyp development up to the point of acquired wnt disrupting mutation when it is seen in crypts with high-grade dysplastic change

With regards to other potential stem cell enriching markers, we demonstrate ectopic crypt lineage tracing from *Sox9*+cells (**Fig 1**) and show that *Olfm4* is expressed in some ectopic crypts in the absence of *Lgr5* expression (**Fig 2**). Additionally, within the new data included in our first revision, we do see very high levels of expression of the putative upper crypt cell marker *Fgfbp1* in ectopic crypts at all lesion stages (**Fig 1**). Given the finding that *Fgfbp1* cells are capable of lineage tracing *in vivo* (2), we suggest that the *Vil1-Grem1* ectopic crypt stem cell population can be identified by these combinations of stem cell markers.

7. In the point-by-point reply, the authors nicely showed (Fig. R6, mentioned only in the table) the potential of villi derived from *vil-Grem1* animals to form organoids. Do they express *Lgr5*? I strongly recommend adding a simplified version of figure (R6) to the manuscript.

Thank you for allowing us to explain why we do not plan to include organoid data in this manuscript. Firstly, the organoid data referred to in the first rebuttal document has been previously published, in Davis *et al.* (ref) Secondly, whilst the study of organoids has many strengths, we feel that this is not the optimal model system for the study of either BMP signalling or cellular plasticity. Organoid growth is dependent on supraphysiological doses of powerful media morphogens (including *Bmp* antagonists and *RSpO*) that together impact homeostatic BMP signalling and generate a state of cellular permissiveness in epithelial cells that we feel could influence and confound the ectopic stem cell behaviour we wish to study. In this situation, we believe that *in vivo* lineage tracing provides a gold standard for proving stem cell functionality, as it allows study of the nuanced intercompartmental signalling balance within the context of the host tissue - a view that is widely supported in the literature (3). This is the reason why we moved from organoid experiments to the more challenging lineage tracing work presented in this current manuscript. Consequently, this work supersedes our previously published work in organoid systems, and we are confident that it stands alone.

The observation that *vil-Grem1* villi give rise to organoids only in ENSW medium is interesting and important. To simplify the figure, one could omit combinations with *Apc-min*. It is known that loss of *Apc* can form ectopic crypts (doi: 10.1126/sciadv.abj0512, doi: 10.1038/s41598-018-38310-y).

We agree with the reviewer that the wnt ligand dependence of ectopic stem cell behaviour is interesting, and we refer to the *in vivo* impact of Porcupine inhibitors (**Fig 5**) which successfully eliminates the ectopic crypts and clearly demonstrates this important point.

7. It would be worthwhile to add representative panels showing individual stainings of Hi-PLEX ISH for *vil-Grem1* (similarly to Fig. S4A) – or at least include the staining as shown in the point-by-point reply Fig. R7 (please indicate the color code for each panel). If possible, please clarify the color code for Fig. 4C in a way that clearly indicates which probes/Abs are used for each

This has been done in new **Figure S4B**, for which we have clearly marked the colour codes for each probe.

These revisions have substantially strengthened the manuscript and fully addressed the reviewers' concerns through extensive additional experimental work. We are confident in the improvements made and submit this revised version for your reconsideration. Please let us know if you need any further information.

Sincerely,

Hayley Belnoue-Davis

Eoghan Mulholland

Simon Leedham

References

1. Shyer AE, Huycke TR, Lee C, Mahadevan L, Tabin CJ. Bending Gradients: How the Intestinal Stem Cell Gets Its Home. *Cell*. 2015 Apr;161(3):569–80.
2. Capdevila C, Miller J, Cheng L, Kornberg A, George JJ, Lee H, et al. Time-resolved fate mapping identifies the intestinal upper crypt zone as an origin of Lgr5+ crypt base columnar cells. *Cell*. 2024 Jun;187(12):3039-3055.e14.
3. Svendsen CN, Sofroniew M V. Lineage tracing: The gold standard to claim direct reprogramming in vivo. *Mol Ther*. 2022 Mar 2;30(3):988–9.